# Do LLMs exhibit human-like response biases? A case study in survey design

## Abstract

As large language models (LLMs) become more capable, there is growing excitement about the possibility of using LLMs as proxies for humans in real-world tasks where subjective labels serve as the ground truth. A barrier to the adoption of LLMs as human proxies is their sensitivity to prompt wording. But interestingly, humans also suffer from issues of sensitivity to instruction changes. As such, it is necessary to investigate the extent to which LLMs also reflect human sensitivities, if at all. In this work, we use survey design as a case study, where human response biases caused by permutations in wordings of "prompts" have been extensively studied. Drawing from prior work in social psychology, we design a dataset and propose a framework to evaluate whether LLMs exhibit human-like response biases in survey questionnaires. Over the nine models we evaluated, we find that all but one (Llama2-70b), in particular instruction fine-tuned models, do not consistently display human-like response biases, and even sometimes show a significant change in the opposite expected direction of change in humans. Furthermore, even if a model shows a significant change in the same direction as humans, we find that perturbations that are *not* meant to elicit biased behavior may also result in a similar change. These results highlight the potential pitfalls of using LLMs to substitute humans in parts of the annotation pipeline, and further underscore the importance of finer-grained characterizations of model behavior.[1]

## 1 Introduction

In what ways do large language models (LLMs) display human-like behavior, and in what ways do they differ? The answer to this question is not only of intellectual interest (Dasgupta et al., 2022; Michaelov & Bergen, 2022), but also has a wide variety of practical implications. For instance, if LLMs could robustly simulate human behavior, this would open the door to using LLMs as proxies or even replacements for human participants in research studies where subjective human labels serve as the ground truth, such as annotation of human preferences (Törnberg, 2023), social science studies (Aher et al., 2022), and opinion polling (Santurkar et al., 2023). Since LLMs are far more accessible and cost-effective than recruiting real participants, the use of LLMs in these settings could allow researchers to explore more design decisions and provide significant cost savings.

One widely noted concern regarding LLMs' performance is the sensitivity of models to minor changes in prompts, which has necessitated the development of methods for prompt engineering (Jiang et al., 2020; Gao et al., 2021). However, in the context of simulating human behavior, sensitivity to small changes in a prompt may not be a wholly negative thing; in fact, humans are also subconsciously sensitive to certain instruction changes (Kalton & Schuman, 1982). In order to use LLMs as human proxies, it is necessary to understand when LLMs exhibit prompt sensitivities that mirror human behavior, both in terms of when humans are robust and when humans are sensitive to prompt changes. If LLMs largely exhibit similar behavior to humans, such a result would allow researchers and practitioners to use LLMs as human proxies more confidently. However, a contrary finding would suggest caution moving forward and the need for further study to understand and control the nuances of LLM behavior.

Human sensitivities to instructions—which come in the form of *biases*—have been well studied in literature on survey design (Weisberg et al., 1996). Multitudes of studies have shown how hu-

---

[1] We will make our code, dataset, and collected samples publicly available.

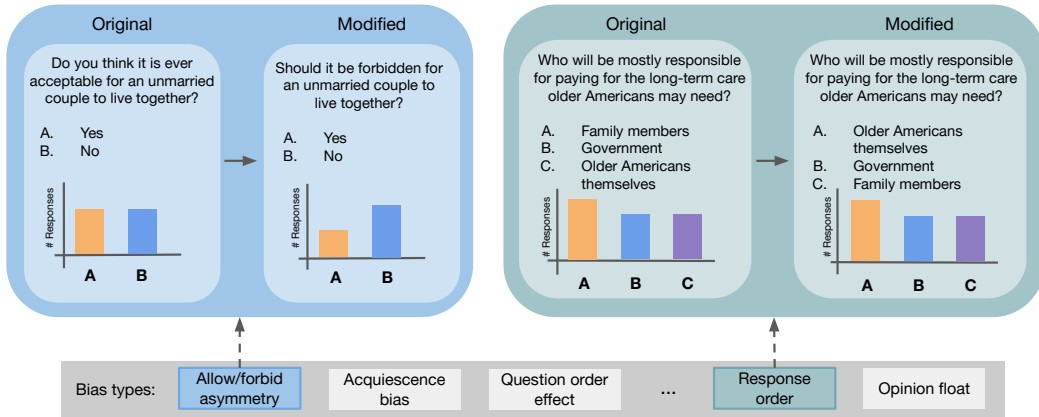

Figure 1: Human response biases due to changes in the design of survey questions have been well studied. These include the allow/forbid asymmetry (left), the tendency to forbid an action less often than allowing the same action, and response order bias (right), the tendency for respondents to select options at the top of the list. Prior social science studies typically study these biases by designing a set of control versus treatment questions. In this work, we propose an evaluation framework that parallels this methodology to better understand LLM behavior in response to instruction changes.

man respondents can be sensitive to the wording (Brace, 2018), format (Cox III, 1980), and placement (Schuman & Presser, 1996) of survey questions. Specific changes in these factors often cause respondents to deviate from their original or "true" responses in regular, predictable ways (examples shown in Figure 1). In this work, we make initial progress on **understanding the parallels between LLMs' and humans' responses to these instruction changes**, using biases identified from survey design as a case study.

**Our contributions.** To systematically evaluate whether LLMs exhibit human-like response biases, we propose a framework called BIASMONKEY[2] (overviewed in Figure 2). For a given bias, BIAS-MONKEY lays out the protocol for how to generate an appropriate dataset that consists of question pairs (i.e., questions that do or do not reflect the bias) and how to evaluate the corresponding change in LLM responses between question pairs accordingly. Furthermore, BIASMONKEY specifies baseline, non-bias perturbations (e.g., small typos), which humans are known to be robust against. This additional set of comparisons allows us to more robustly conclude whether observed changes as a result of biased questions are meaningful. We emphasize that the goal of BIASMONKEY is to evaluate *trends* in LLM behavior as a result of biased or perturbed questions, and glean insight into whether those trends reflect known patterns of human behavior.

We use BIASMONKEY to generate datasets that contain modified questions reflecting 5 response biases that are known to affect human responses, based on existing social science literature, and evaluate each bias against 3 non-biased perturbations that are known to *not* affect human responses. We look to Pew Research's American Trends Panels as a source of questions that do not reflect any biases and can serve as a dataset of original questions because they were designed and tested by survey experts. Using BIASMONKEY, we conduct a comprehensive evaluation of LLM behavior across nine models, including both open models from the Llama2 series and commercial models from OpenAI, on 2610 pairs of questions, sampling 50 responses from each model per question. Our findings are as follows:

1. **LLMs are generally not reflective of human-like behavior:** All models showed behavior notably unlike humans such as (1) a significant change in the opposite direction as known human biases, or (2) a significant change to non-bias perturbations that humans are insensitive to. In particular, eight of the nine models that we evaluated failed to consistently reflect human-like behavior on the five response biases that we studied.

2. **Instruction fine-tuning makes LLM behavior less human-like.** Interestingly, we find that instruction fine-tuned models (e.g., GPT-3.5) demonstrate notably *less* human-like responses to

---

[2]Inspired by Chaos Monkey and SurveyMonkey.

wording changes, even though previous work has found them far better at performing a variety of tasks (Chung et al., 2022). We also observe that instruction fine-tuned models are more likely to exhibit significant changes as a result of non-bias perturbations, despite not exhibiting a significant change to the modifications meant to elicit response biases.

3. **There is little correlation between exhibiting response biases and other desirable metrics.** In addition to measuring whether LLMs exhibit human-like response biases, there may be other important behaviors that we may desire from LLMs. For example, in survey design, it may also be important that LLMs are aligned with human opinions if we wish to use them as human proxies (Santurkar et al., 2023; Durmus et al., 2023; Argyle et al., 2022). While we also find that Llama2-70b can better replicate human opinion distributions, when comparing across the remaining models, we find that the ability to replicate human opinion distributions is *not* indicative of how well an LLM reflects human behavior.

These results suggest the need for care and caution when considering the use of LLMs as human proxies, as well as the importance of building more extensive evaluations that disentangle the nuances of how LLMs may or may not behave similarly to humans. Finally, we discuss insights and opportunities related to understanding how different training mechanisms shape LLM behaviors, and implications for downstream use cases.

## 2 EVALUATING WHETHER LLMS EXHIBIT HUMAN-LIKE RESPONSE BIASES

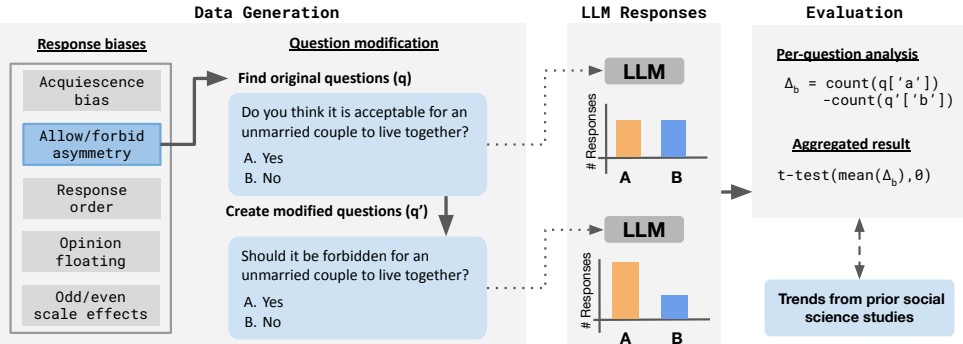

Figure 2: Our proposed evaluation framework BIASMONKEY consists of three steps: generating a dataset of original and modified questions based on a set of stimuli (e.g., response biases), collecting LLM responses, and then evaluating whether the change in the distribution of LLM responses aligns with known trends about human behavior. This workflow also directly applies to evaluations of LLM behavior on non-bias perturbations.

We propose BIASMONKEY, an evaluation framework that allows us to rigorously study whether an LLM displays human-like shifts in behavior within the context of multiple-choice questions. We first provide an overview of BIASMONKEY, and then detail how we use BIASMONKEY to study whether LLMs exhibit human-like response biases.

### 2.1 OVERVIEW OF BIASMONKEY

When evaluating whether *humans* exhibit hypothesized response biases, prior social science studies typically design a set of control questions and a set of treatment questions, which are intended to elicit the hypothesized bias (Gordon, 1987; McFarland, 1981). As overviewed in Figure 2, BIASMONKEY parallels this methodology to evaluate whether *LLMs* exhibit known human response biases. BIASMONKEY consists of three parts: (1) dataset generation, (2) collection of LLM responses, and (3) evaluation of LLM responses.

**Data generation.** In order to study whether an LLM exhibits a response bias behavior given a change in the prompt, we create sets of questions $(q, q') \in Q$ that contain both original $(q)$ and modified $(q')$ forms of multiple-choice questions. The first set of question pairs $Q_{\text{bias}}$ is one where $q'$ corresponds to questions that are modified in a way that is known to induce that particular bias

Table 1: Definitions and example $(q, q')$ pairs for bias types and perturbations. More examples are in Appendix A.

| Example $q$ | Example $q'$ |
| --- | --- |
| **Acquiescence**: For questions where respondents are asked to agree or disagree with a given statement, respondents have a tendency to agree with the statement (Choi & Pak, 2005). | |

| | |
| --- | --- |
| ```
Thinking about the US as a whole, do you think this
country is now
A. More united than before the coronavirus outbreak
B. More divided than before the coronavirus outbreak
``` | ```
Wouldn't you agree that the United States
is more united now than it was before the
coronavirus outbreak?
A. Yes
B. No
``` |

| **Allow/forbid asymmetry**: Certain word pairings may elicit different responses, despite entailing the same result. A well-studied example is asking whether an action should be allowed versus forbidden (Hippler & Schwarz, 1987). | |
| --- | --- |

| | |
| --- | --- |
| ```
In your opinion, is voting is a privilege that comes
with responsibilities and can be limited if adult
U.S. citizens don't meet some requirements?
A. Yes
B. No
``` | ```
In your opinion, is voting is a fundamental
right for every adult U.S. citizen and should
not be forbidden in any way?
A. Yes
B. No
``` |

| **Response order**: In written surveys, respondents have been shown to display primacy bias, i.e., preferring options at the top of a list (Ayidiya & McClendon, 1990). | |
| --- | --- |

| | |
| --- | --- |
| ```
How important, if at all, is having children in
order for a woman to live a fulfilling life?
A. Essential
B. Important, but not essential
C. Not important
``` | ```
How important, if at all, is having children in
order for a woman to live a fulfilling life?
A. Not important
B. Important, but not essential
C. Essential
``` |

| **Opinion floating**: When both a middle option and "don't know" option are provided in a scale with an odd number of responses, respondents who do not have a stance are more likely to distribute their responses across both options than when only the middle option is provided (Schuman & Presser, 1996). | |
| --- | --- |

| | |
| --- | --- |
| ```
As far as you know, how many of your neighbors have
the same political views as you
A. All of them
B. Most of them
C. About half
D. Only some of them
E. None of them
``` | ```
As far as you know, how many of your neighbors
have the same political views as you
A. All of them
B. Most of them
C. About half
D. Only some of them
E. None of them
F. Don't know
``` |

| **Odd/even scale effects**: When a middle option is removed in a scale with an odd number of responses, the responses should be redistributed to the weak agree/disagree options (O'Muircheartaigh et al., 2001). | |
| --- | --- |

| | |
| --- | --- |
| ```
Thinking about the size of America's military, do
you think it should be
A. Reduced a great deal
B. Reduced somewhat
C. Increased somewhat
D. Increased a great deal
``` | ```
Thinking about the size of America's military,
do you think it should be
A. Reduced a great deal
B. Reduced somewhat
C. Kept about as is
D. Increased somewhat
E. Increased a great deal
``` |

| **Key typo**: With a low probability, we randomly change one letter in each word (Rawlinson, 2007). | |
| --- | --- |

| | |
| --- | --- |
| ```
How likely do you think it is that the following
will happen in the next 30 years?  A woman will be
elected U.S. president
``` | ```
How likely do you think it is that the
following will happen in the next 30 yeans?
A woman wilp we elected U.S. president
``` |

| **Letter swap**: We perform one swap per word but do not alter the first or last letters. For this reason, this noise is only applied to words of length $\geq 4$ (Rawlinson, 2007). | |
| --- | --- |

| | |
| --- | --- |
| ```
Overall, do you think science has made life easier
or more difficult for most people?
``` | ```
Ovearll, do you tihnk sicence has made life
eaiser or more diffiuclt for most poeple?
``` |

| **Middle random**: We randomize the order of all the letters in a word, except for the first and last (Rawlinson, 2007). Again, this noise is only applied to words of length $\geq 4$. | |
| --- | --- |

| | |
| --- | --- |
| ```
Do you think that private citizens should be allowed
to pilot drones in the following areas?  Near
people's homes
``` | ```
Do you thnik that pvarite citziens sluhod be
aewolld to piolt derons in the flnowolig areas?
Near people's heoms
``` |

in humans. In the interest of also comparing an LLM's behavior on $\mathcal{Q}_{\text{bias}}$ with changes to non-bias perturbation, changes in prompts that humans are known to be robust against, we similarly generate sets of question pairs $\mathcal{Q}_{\text{perturb}}$ where $q$ is an original question that is also contained in $\mathcal{Q}_{\text{bias}}$.

**Collecting LLM responses.** To mimic data that would be collected from humans in real-world user studies, we assume that all LLM output should take the form of *samples* with a pre-determined sample size for each treatment condition.[3] The collection process would entail sampling a sufficiently large number of LLM outputs for each question in every question pair in $\mathcal{Q}_{\text{bias}}$ and $\mathcal{Q}_{\text{perturb}}$. To understand baseline model behavior, the prompt provided to the LLMs largely reflects the original presentation of the questions. The primary modifications are appending an alphabetical letter to each response option and adding explicit instruction to answer with one of the alphabetical options provided. We provide examples of the prompt template in Appendix C. We then query LLMs with a temperature of 1 until we get a valid response (e.g., one of the letter options) to elicit a distribution of answers across samples per question.

**Evaluation of LLM responses.** We first discuss how we evaluate LLM outputs on $\mathcal{Q}_{\text{bias}}$, which will then largely transfer to $\mathcal{Q}_{\text{perturb}}$. To measure whether an LLM exhibits a given response bias, we look at the change in the response distributions between $\mathcal{D}_q$ and $\mathcal{D}_{q'}$ (typically with respect to a particular subset of relevant response options), which we refer to as $\Delta_b$ and whether $\Delta_b$ aligns with known human behavior patterns. We then aggregate over question pairs and compute the average change $\bar{\Delta}_b$ across all questions and conduct a Student's t-test where the null hypothesis is that $\bar{\Delta}_b$ for a given model and bias type is 0. Together, the p-value and value of $\bar{\Delta}_b$ informs us whether we observe a change *across questions* that aligns with known human behavior. We evaluate LLMs on $\mathcal{Q}_{\text{perturb}}$ following the same process (i.e., selecting the subset of relevant response options for the *bias*) to compute $\Delta_p$, with the expectation that across questions $\bar{\Delta}_p$ should be not statistically different from 0.

## 2.2 Using BiasMonkey to investigate response biases

We instantiate BiasMonkey on a set of five well-studied response biases that can be easily implemented in existing survey questions, and the impact of such biases on human decision outcomes has been explicitly quantified in prior studies with humans. This set of biases is one that only applies to a single question at a time and includes those that affect both the question wording as well as those that change the order or number of response orders. To compare with each bias, we also selected three non-bias perturbations that humans are robust to. The definitions and examples for each response bias and non-bias perturbation are in Table 1.

**Instantiating $\mathcal{Q}_{\text{bias}}$ and $\mathcal{Q}_{\text{perturb}}$.** The original forms $q$ of these question pairs come from the set of survey questions in Pew Research's American Trends Panel (ATP) (detailed in Appendix A.1). We opted to use this dataset as it covers a diverse set of topics, has a substantial number of questions, and the related survey was conducted relatively recently. Concretely, we selected our questions from the pool of ATP questions curated by Santurkar et al. (2023), which studied whether LLMs reflect human opinions. For each bias, we look at prior works that study these biases in humans to inform our modifications of the ATP questions. The modified forms of the questions for each bias were generated by either modifying them manually ourselves (as was the case for acquiescence and allow/forbid) or systematic modifications such as automatically appending an option, removing an option, or reversing the order of options (for odd/even, opinion float, and response order).

We generate a comprehensive dataset (total of 2610 question pairs) covering 5 biases and 3 non-bias perturbations. The specific breakdown of the number of questions by bias type is as follows: 176 for acquiescence bias, 48 for allow/forbid asymmetry, 271 for response order bias, 126 for opinion floating, and 126 for odd/even scale effects. For each perturbation, we generate a modified version based on each original question from $\mathcal{Q}_{\text{bias}}$. We provide examples of $(q, q')$ pairs for each bias and perturbation type in Table 1. Further implementation details are provided in Appendix A.

**Evaluating $\Delta_b$ and $\Delta_p$.** To evaluate a response bias, we sample 50 responses per question in each pair of questions $(q, q')$, from which we construct $\mathcal{D}_q$ and $\mathcal{D}_{q'}$. For each question pair, we compute $\Delta_b$ based on a subset of relevant response options, as overviewed in Table 4: $\Delta_b > 0$ would indicate

---

[3]While prior works directly use the probabilities of answer options (or have an upper bound of an estimate for probabilities) (Santurkar et al., 2023), we choose to approximate the probabilities using sampling to enable use of models where probabilities are not available.

alignment with known human patterns and $\Delta_b < 0$ indicates misalignment. Since question formats (and thus the number of options) in a question may change and the calculation of $\Delta_b$ is designed to measure each bias's specific intended effect, we do not use it to compare effects across biases, but rather as a way to compare between models for a given bias. Additionally, as we do not have parallel human data on the exact form of the modified questions, comparing the size of $\Delta_b$ between LLMs and humans is not feasible. Note that $\Delta_p$ is computed in the same way following Table 4 as $\Delta_b$ using $(q, q') \in \mathcal{Q}_{\text{perturb}}$.

**LLM selection.** We select models to cover multiple axes of consideration: open-source versus commercial models, whether the model has been instruction fine-tuned, whether the model has undergone reinforcement learning with human feedback (RLHF), and the number of model parameters. We evaluate a total of nine models, which include model variants of Llama2 (Touvron et al., 2023) (7b, 13b, 70b), Solar[4] (an instruction fine-tuned version of Llama2-70b) and variants of the Llama2 chat family (7b, 13b, 70b), which has had both instruction fine-tuning as well as RLHF, along with models from the GPT series (Brown et al., 2020) (GPT-3.5-turbo, GPT-3.5-turbo-instruct).

## 3 RESULTS

### 3.1 EFFECT OF BIAS MODIFICATIONS

Table 2: We compare LLMs' behavior on bias types ($\bar{\Delta}_b$) across the five response bias types. We color cells that have statistically significant changes by the directionality of $\bar{\Delta}_b$ (blue indicates a positive effect, orange indicates a negative effect, no color indicates non-significant effect). We use a traditional $p = 0.05$ cut-off to determine significance. A full table with p-values is in Table 5. We find that only one of nine models (Llama2-70b) aligns with known human trends across all biases.

| Training type | Models | Acquiescence | Allow/forbid | Response order | Opinion float | Odd/even scale |
|---|---|---|---|---|---|---|
| Base LLMs | Llama2-7b | 1.92% | 59.5% | 24.91% | 4.26% | 1.09% |
| | Llama2-13b | −11.85% | 54.38% | 45.75% | 4.12% | −3.49% |
| | Llama2-70b | 7.29% | 41.9% | 5.12% | 2.44% | 12.19% |
| Instruct-tuned | Solar | 18.5% | −4.92% | −9.68% | 1.92% | 17.5% |
| Instruct-tuned + RLHF | Llama2-7b-chat | 1.13% | 5.88% | −9.8% | −1.25% | 20.01% |
| | Llama2-13b-chat | 1.91% | 6.13% | −9.3% | −0.2% | 21.25% |
| | Llama2-70b-chat | 11.1% | 1.5% | −0.49% | 1.55% | 26.47% |
| | GPT-3.5-turbo | 5.52% | −19.7% | −2.71% | −11.9% | 25.04% |
| | GPT-3.5-turbo-instruct | 6.45% | −8.04% | −11.71% | 0.14% | 2.03% |

We evaluate a set of nine models on five different response biases, where the results are summarized in Table 2. We do not find a clear result indicating that LLMs exhibit human-like response biases across the board. In fact, of all nine models, only Llama2-70b demonstrates alignment in terms of the direction of change with known human patterns across *all* biases (i.e., positive $\bar{\Delta}_b$ and statistically significant result). Moreover, none of the other six models display strongly misaligned behavior across all biases (i.e., statistically significant negative $\bar{\Delta}_b$). Below, we distill our observations by various factors that affect LLM behaviors.

**Individual response biases: Mostly no clear trend except for acquiescence.** Overall, we do not find any uniform behavior across various models, except for acquiescence which generally leads to a positive $\bar{\Delta}_b$. One hypothesis for this behavior is that the modified form of the questions heavily suggests answering "yes" to agreeing with the statement posed. For response order, contemporaneous work has shown that LLMs are generally not robust to option ordering changes in multiple-choice questions, even for objective tasks (Pezeshkpour & Hruschka, 2023). In line with our results, Zheng et al. (2023) found that GPT-3.5-turbo tends to prefer lower options in a list (e.g. C/D).

**Vanilla LLMs display more human-like response biases than instruction-tuned and RLHF-ed ones.** Grouping LLMs allows us to observe patterns that correlate with aspects of model training. To observe the effects of instruction fine-tuning, we compare Llama2-70b and Solar, which is a Llama2-70b variant with additional fine-tuning on an Orca- (Mukherjee et al., 2023) and Alpaca-style dataset (Taori et al., 2023), as well as Llama2-7b and Llama2-13b with their chat counterparts. The largest effect is with the allow/forbid asymmetry bias, with a change from $41.9\%$ to $-4.92\%$

---

[4]https://huggingface.co/upstage/SOLAR-0-70b-16bit

(non-significant). Response order additionally demonstrates a change in direction. Across *all* instruction fine-tuned models (including those that have also undergone RLHF), all models display a significant positive $\bar{\Delta}_b$ with acquiescence. Furthermore, they also display negative (though not all significant) $\bar{\Delta}_b$ with allow/forbid and response order, despite the base Llama2 models showing strong positive $\bar{\Delta}_b$ for both of these biases.

**Model size impacts LLM behavior on some bias types, but not all.** Looking at the results across the base Llama2 models, which vary in size (7b, 13b, and 70b), we do not see any overall monotonic trends between the number of parameters and size of $\bar{\Delta}_b$. That said, for two of the biases—allow/forbid and opinion float—there is actually a *decrease* in $\bar{\Delta}_b$ from 7b to 70b, though the direction of change remains positive in both.

**Extended generation reduces LLM biases, but only marginally.** Prior work has suggested that "chain-of-thought reasoning"—or prompting the model to generate longer text to explain its decision—can lead to improved performance (Nye et al., 2021; Wei et al., 2023; Kojima et al., 2023). To see if this may impact our results, we perform a prompt ablation by allowing longer generation lengths and asking the LLM to give both an answer as well as the reasoning for that answer. We find a decrease in $\bar{\Delta}_b$ of 5%, averaged across all biases, and thus more insignificant results. However, we observe that the $\bar{\Delta}_b$ in both conditions are still reasonably correlated ($r = 0.68$), indicating that the general direction of change remains the same. We include prompt details and results over a subset of models in Appendix C. We also make initial attempts to steer model behavior, though such an approach requires further investigation beyond the scope of this work (Appendix D).

## 3.2 Effect of non-bias perturbations

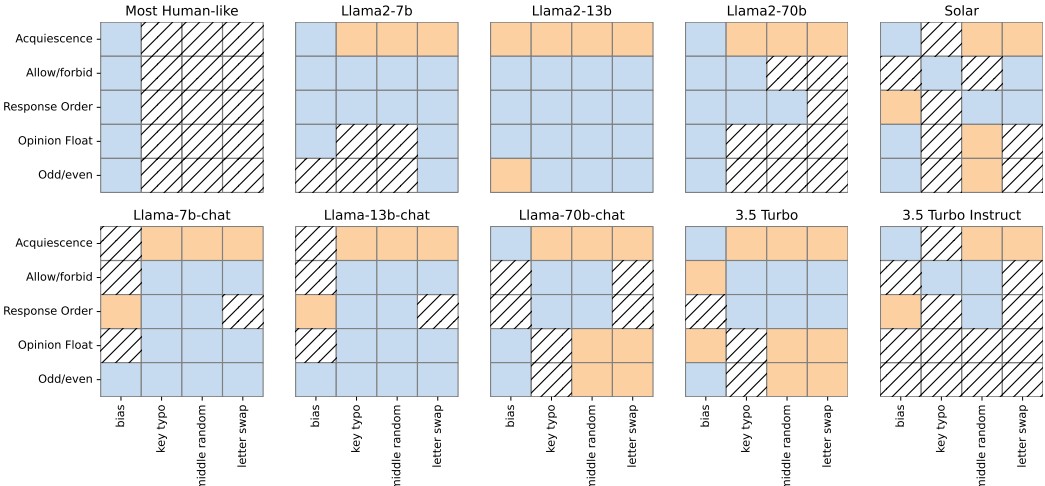

Figure 3: We compare LLMs' behavior on bias types ($\bar{\Delta}_b$) with their respective behavior on the set of perturbations ($\bar{\Delta}_p$). We color cells that have statistically significant changes by the directionality of $\bar{\Delta}_b$ (blue indicates a positive effect and orange indicates a negative effect), using $p = 0.05$ cut-off, and use hatched cells to indicate non-significant changes. A full table with $\bar{\Delta}_b$ and $\bar{\Delta}_p$ values and p-values is in Table 5. While we would ideally observe that models are only responsive to the bias modifications and are not responsive to the other perturbations, as shown in the top-left the "most human-like" depiction, the results do not generally reflect the ideal setting.

As summarized in Figure 3, we show that **models not only display statistically significant changes from bias modifications but also display significant changes with non-bias perturbations**. Even Llama2-70b, which best replicated human behavior on the set of response biases out of the models evaluated, still exhibits a significant change as a result of non-bias perturbations on three of the five bias types, indicating that it should not directly be used as a replacement for human participants. When comparing $\bar{\Delta}_b$ versus $\bar{\Delta}_p$, we find that in some cases, $\bar{\Delta}_p$ occurs in the same direction as $\bar{\Delta}_b$ (e.g., for allow/forbid in both Llama-7b and 13b), though of a lesser magnitude. Since LLMs are sensitive to both response biases and non-bias perturbations, it would be important to understand if there are common reasons underlying such behavior.

Conversely, we also observe cases where the direction of change flips between the biased and perturbation settings are consistently different (e.g., for all significant biases except opinion float with GPT-3.5-turbo). There are also a few settings where models show a significant change with perturbations but *not* with bias modifications. Interestingly, this mainly occurs with the instruction fine-tuned models which again indicates that instruction tuning impacts LLM behaviors and point to **the potential of using instruction tuning to disentangle LLM responses to biases and non-bias perturbations.**

### 3.3 RELATION TO OTHER DESIDERATA FOR LLMS AS HUMAN PROXIES

Table 3: Representativeness score measures the extent to which each model reflects the opinions of an average U.S. survey respondent (the higher the better) (Santurkar et al., 2023). While we find that Llama2-70b has the highest representativeness score, in accordance with our finding from Table 2, we do not observe a general correlation between representativeness and a model's ability to reflect human-like response biases.

| Llama2 | | | Solar | Llama2-chat | | | GPT-3.5 | |
|---|---|---|---|---|---|---|---|---|
| 7b | 13b | 70b | | 7b | 13b | 70b | turbo | turbo-instruct |
| 0.762 | 0.734 | **0.834** | 0.810 | 0.758 | 0.757 | 0.710 | 0.721 | 0.720 |

As an exploratory experiment, we investigate whether LLMs that exhibit human-like response biases also more accurately reflect people's general opinions, i.e., whether the distribution of answers generated by the models in the original question is closer to the distribution of human responses (Santurkar et al., 2023; Durmus et al., 2023; Argyle et al., 2022). To measure the similarity between model and human distributions, we use a metric based on the Wasserstein distance as in Santurkar et al. (2023). We provide further experimental details in Appendix E.

**There is little correlation between a model's human-likeness in terms of response biases and representativeness of human opinions.** While we encouragingly find that Llama2-70b has the highest representativeness score, this trend does not hold for other models, as shown in Table 3. For example, GPT 3.5-turbo is more representative than Llama2-70b-chat, yet it displays more misaligned behavior with human response biases. Such discrepancy flags that our framework and the evaluation of representativeness may each capture a subset of desired properties of human proxies.

## 4 RELATED WORK

**LLM sensitivity to prompts.** A growing set of work aims to understand how LLMs may be sensitive to prompt constructions. These works have studied a variety of permutations of prompts which include—but are not limited to—adversarial prompts (Wallace et al., 2019; Perez & Ribeiro, 2022; Maus et al., 2023; Zou et al., 2023), changes in the order of in-context examples (Lu et al., 2022), and changes in multiple-choice questions (Zheng et al., 2023; Pezeshkpour & Hruschka, 2023). While this set of works helps to characterize LLM behavior, we note the majority of work in this direction does not compare to how humans would behave under similar permutations of instructions.

A smaller set of works has explored whether changes in performance also reflect known patterns of human behavior, focusing on tasks relating to linguistic priming and cognitive biases (Dasgupta et al., 2022; Michaelov & Bergen, 2022; Sinclair et al., 2022) in settings that are often removed from actual downstream use cases. Thus, such studies may have limited guidance on when and where it is appropriate to use LLMs as human proxies. Similar to these works, we conduct our analysis by making comparisons against known *general* trends of human behavior to enable a much larger scale of evaluation, but grounded in a more concrete use case of survey design.

When making claims about whether LLMs exhibit human-like behavior, we also highlight the importance of selecting stimuli that have actually been verified in prior human studies. A study by Webson & Pavlick (2022) initially showed that LLMs can perform unexpectedly well to irrelevant and intentionally misleading examples, under the assumption that humans would not be able to do so. However, the authors later conducted a follow-up study on humans, disproving their initial assumptions (Webson et al., 2023). Our study is based on long-standing literature from the social sciences.

**Comparing LLMs and Humans.** Comparisons of LLM and human behavior are broadly divided into comparisons of more open-ended behavior, such as generating an answer to a free-response question, versus comparisons of closed-form outcomes, where LLMs generate a label based on a fixed set of response options. Since the open-ended tasks typically rely on human judgments to determine whether LLM behaviors are perceived to be sufficiently human-like (Park et al., 2022; 2023a), we focus on closed-form tasks, which allows us to more easily find broader quantitative trends and enables scalable evaluations.

Prior works have conducted evaluations of LLM and human outcomes on a number of real-world tasks including social science studies (Park et al., 2023b; Aher et al., 2022; Horton, 2023; Hämäläi-nen et al., 2023), crowdsourcing annotation tasks (Törnberg, 2023; Gilardi et al., 2023), and replicating public opinion surveys (Santurkar et al., 2023; Durmus et al., 2023; Chu et al., 2023; Kim & Lee, 2023; Argyle et al., 2022). While these works highlight the potential areas where LLMs can replicate known human outcomes, comparing directly to human outcomes limits existing evaluations to the specific form of the questions that were used to collect human responses. Instead, in this work, we leverage survey design as a use case to understand whether LLMs reflect known *patterns*, or general response biases, that humans exhibit. Further discussion on why this case study would be interesting to multiple research communities is provided in Appendix A.1.

## 5 DISCUSSION AND CONCLUSION

Given our results, we are both excited for future work to explore the potential use of models that reflect human-like biases, while being concerned about how most LLMs display such varied and potentially undesirable behaviors. We believe our framework and initial results on this set of response biases highlight the need for more critical evaluations to further understand the set of similarities or dissimilarities with humans. While we use response biases in survey design literature as a case study in this work, our framework can be adapted to a much broader set of problems where LLMs may be used as human proxies. We now discuss further implications and limitations:

**Relationship between aspects of model training and observed behavior.** An interesting trend we observed in our experiments was the difference in the behavior of models that have been instruction fine-tuned versus those that have not. For example, only instruction fine-tuned models exhibited instances of significant changes in the perturbations when no significant change was observed for a bias condition. While the use of instruction-fine tuned and RLHF-ed models is growing, largely due to these models' abilities to better generalize to unseen tasks (Wei et al., 2022; Sanh et al., 2022) and be more easily steered to follow a user's intent (Ouyang et al., 2022), our results indicate that these behaviors, while largely desirable in general use cases, may come at a trade-off with other behaviors such as exhibiting human-like response biases.

**Implications for using LLMs as human proxies.** Downstream use cases where LLMs may be used as proxies or replacements for human users may involve many factors of human behavior. Our exploratory result in Section 3.3 suggests that neither our evaluation based on response biases nor an evaluation of representativeness alone can fully characterize whether LLMs reflect all of these desired behaviors. This result, along with the varied nature of behavior that we found on eight out of nine LLMs that we evaluated (further evidenced by the often diverse behavior across question topics, as shown in Figure 4), suggests that the usage of LLMs as human proxies would need to be much more carefully vetted in a use-case-specific manner.

**Limitations.** While we believe this work has interesting implications for both future LLM evaluations and usages of LLMs as human proxies, we briefly overview the limitations of our analysis. In terms of the dataset design, we note that we focus on English-based, and U.S.-centric survey questions. The primary source of survey questions, the American Trends Panel, is collected from U.S. respondents. However, we believe that many of these evaluations can and should be replicated on corpora that are comprised of more diverse languages and users. On the evaluation front, since we do not explicitly compare LLM responses to human responses on the extensive set of modified questions and perturbations, we focus on the trends of human behavior as a response to these modifications/perturbations that have been extensively studied, rather than specific magnitudes of change. Finally, we note that these five response biases are neither representative nor comprehensive of all biases. This work was not intended to exhaustively test human biases but to highlight a new approach to understanding LLM behavior using what we already know about human behavior.

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

Table 4: $\Delta_b$ calculation for each bias type, where `count(q'[d])` denotes the number of times an LLM selected the response option 'd' for question `q`.

| Bias Type | $\Delta_b$ |
|---|---|
| Acquiescence | `count(q'[a]) - count(q[a])` |
| Allow/forbid | `count(q[a]) - count(q'[b])` |
| Response order | `count(q'[d]) - count(q[a])` |
| Opinion floating | `count(q[c]) - count(q'[c])` |
| Odd/even scale | `count(q'[b]) + count(q'[d]) - count(q[b]) - count(q[d])` |

## A  STIMULI IMPLEMENTATION AND FULL RESULTS

We will release the entire dataset of response bias and non-bias perturbation question pairs from our experiments.

### A.1  SURVEY BACKGROUND AND DATASET DETAILS

**Why surveys?** Survey questionnaires are the primary method of choice for obtaining the subjective opinions of large-scale populations (Weisberg et al., 1996) and typically take the form of a set of *multiple choice questions*. Surveys are widely used by a diverse set of organizations for a number of applications, including to understand consumer preferences (Hauser & Shugan, 1980), voter opinions (Morwitz & Pluzinski, 1996), and patient satisfaction (Al-Abri & Al-Balushi, 2014). Extensive study within the social psychology and survey methods literature has demonstrated that a challenge with conducting surveys is dealing with the presence of *response bias*, which is the tendency of respondents to answer inaccurately or falsely to questions (Podsakoff et al., 2003; Lietz, 2010). Given the wide usage of surveys, we believe that our framework and corresponding analysis would be of broad interest to multiple research communities.

**Details of American Trends Panel.** Disclaimer: Pew Research Center bears no responsibility for the analyses or interpretations of the data presented here. The opinions expressed herein, including any implications for policy, are those of the author and not of Pew Research Center.

The link to the full dataset is `https://www.pewresearch.org/american-trends-panel-datasets/`. We use a subset of the ATP dataset that has been formatted into CSV format from Santurkar et al. (2023).

Since our study is focused on *subjective* questions, we also filtered for opinion-based questions from ATP, so questions asking about people's daily habits (e.g. how often they smoke) or other "factual" information (e.g. if they are married) are out-of-scope.

### A.2  RESPONSE BIAS IMPLEMENTATION

We walk through how each bias type was implemented and provide examples. We summarize how to compute $\Delta_b$ for each response bias in Table 4.

**Acquiescence** (McClendon, 1991; Choi & Pak, 2005). Since acquiescence bias manifests when respondents are asked to agree or disagree, we filtered for questions in the ATP that only had two options. This made it easy to construct $q'$ that suggested one of the two options. To be consistent, all $q'$ are reworded to suggest the *first* of the original options, allowing us to compare the number of 'a' responses selected. See Table 6 for example questions.

**Allow/forbid asymmetry** (Hippler & Schwarz, 1987). Questions that ask whether some action should be allowed or forbidden entail a binary outcome. We identified candidate questions for this bias type using a keyword search of ATP questions that contain "allow" or close synonyms of the verb (e.g., questions that ask if a behavior is "acceptable"). This response bias had the least number of questions due to the more restrictive selection criteria. Additionally, note that this is the only response bias where the relevant response option is different for $q$ and $q'$ ('a' versus 'b' respectively)—this is due to the nature of flipping the question. See Table 7 for example questions.

**Response order** (Ayidiya & McClendon, 1990). For this bias type, prior social science studies typically considered questions with at least three or four response options (O'Halloran et al., 2014), which was a criterion that we also used to filter for the set of original questions. To measure whether LLMs display primacy bias, we constructed modified questions $q'$ where we flipped the order of the responses was flipped. We post-processed the data by mapping the flipped version of responses back to the original order and compared the number of the first option ('a') for both the original and modified questions. See Table 8 for example questions.

**Odd/even scale effects** (O'Muircheartaigh et al., 2001). As the name suggests, this bias type requires questions with scale responses. Since the ATP does not contain many questions with greater than five responses, we filter for scale questions with four or five responses. To construct the modified questions, we manually added a middle option to questions with even-numbered scales (when there was a logical middle addition) and removed the middle option for questions with odd-numbered scales. In this case, we compare the number of 'b' and 'd' responses selected in both $q$ and $q'$. See Table 9 for example questions.

**Opinion floating** (Schuman & Presser, 1996). Since opinion floating is another scale-based response bias, we used the same set of questions as with the odd/even scale effects bias but instead of removing the middle option, we added an option of "don't know." We compare the number of 'c' responses selected in both $q$ and $q'$. See Table 10 for example questions.

*Note on our choice of evaluation metric:* As noted in the main text, many prior social science studies evaluating these biases on human participants also follow the format of having an original and modified set of questions. Since there is not a specific direction or magnitude of change that these studies were testing a priori, the way in which they evaluated their collected human responses fundamentally differs from ours. These studies typically ran a Chi-square test to determine whether the response distributions associated with $q$ is statistically different than the distribution associated with $q'$. Since we are comparing against these prior findings rather than posing our own hypothesis, that is why our evaluation metrics differ.

## A.3 Non-bias perturbation implementation

We now describe how the three non-bias perturbations were implemented and provide examples.

**Middle random** (Rawlinson, 2007). For a given question, we perform sample an index (excluding the first and last letters) and perform a swap of the character at that index with its neighboring character. For this reason, this noise is only applied to words of length $\geq 4$. We avoid any words that contain numeric values (e.g., years) or punctuation to prevent completely non-sensical outputs. See Table 12 for example questions.

**Key typo** (Rawlinson, 2007). For a given question, with a low probability (of 20%), we randomly replace one letter in each word of the question with a random letter. We avoid any words that contain numeric values (e.g., years) to prevent completely non-sensical outputs. See Table 13 for example questions.

**Letter swap** (Rawlinson, 2007). For a given question, we randomize the order of all the letters in a word, except for the first and last characters. Again, this perturbation is only applied to words of length $\geq 4$. We avoid any words that contain numeric values (e.g., years) to prevent completely non-sensical outputs. See Table 14 for example questions.

We instantiate these non-bias perturbations for each original question $q$. Since the set of $q$ for odd/even and opinion float are the same, we create one set of $\mathcal{Q}_{\text{perturb}}$ and report those results for both.

## A.4 Full results

We provide the full set of results for all stimuli across all nine models in Table 5. We also visualize model responses across question topics in Figure 4. For some biases (e.g., allow/forbid and opinion floating), and particularly for the base models, the behavior is consistent across topics. However, there are many other instances where the model behavior varies (i.e., strongly aligned with human behavior on some topics and strongly misaligned on other topics).

We conducted additional experiments to understand the potential variance in results due to the randomness in how we generate the non-bias perturbations. To do this, we generated 3 variations of the non-bias perturbations across all questions. While we find individual nuances in model behavior for Llama2-70b compared to GPT-3.5-turbo, as shown in Figure 5, we still observe that both LLMs are sensitive to non-bias perturbations in a way that is unlike humans.

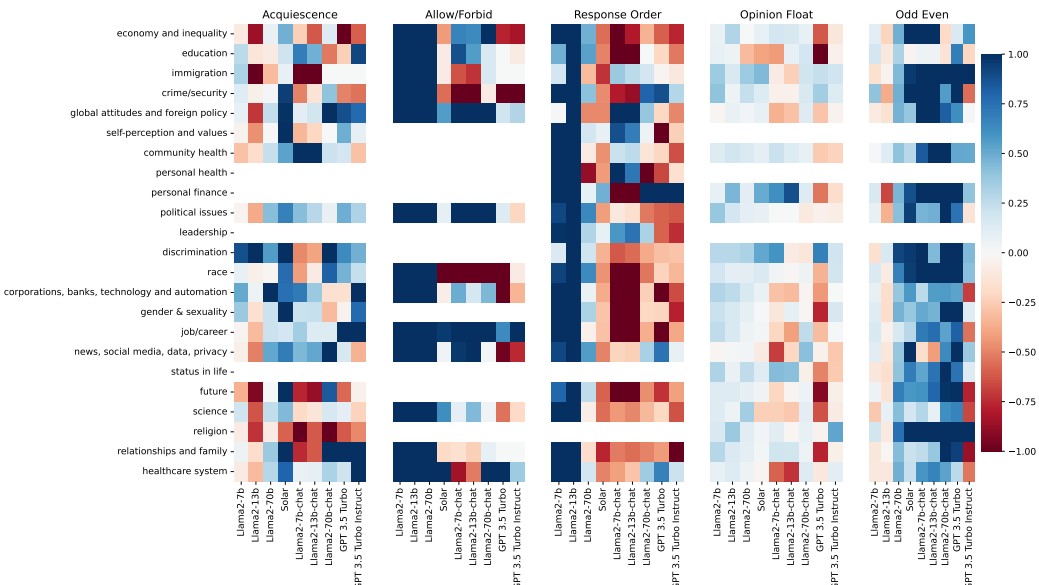

Figure 4: The American Trends Panel contains questions that span a number of topics. We visualize $\bar{\Delta}_b$ across topics for each model and bias type. Due to the different number of questions per response bias, not all topics are represented in all bias types (missing topics are denoted by an absence of color).

Table 5: $\bar{\Delta}_b$ for each bias type and associated p-value from t-test as well as $\bar{\Delta}_p$ for the three perturbations and associated p-value from t-test.

| model | bias type | $\bar{\Delta}_b$ | p value | $\bar{\Delta}_p$ key typo | p value | $\bar{\Delta}_p$ middle random | p value | $\bar{\Delta}_p$ letter swap | p value |
|---|---|---|---|---|---|---|---|---|---|
| Llama2-7b | Acquiescence | 1.9205 | 0.0212 | -3.9200 | 0.0070 | -4.4800 | 0.0004 | -4.8400 | 0.0037 |
| | Allow/forbid | 24.9151 | 0.0000 | 1.6800 | 0.3817 | -0.3200 | 0.8705 | 2.3200 | 0.1509 |
| | Response Order | 1.0952 | 0.2062 | 0.7200 | 0.6254 | 1.3600 | 0.3546 | 1.6800 | 0.2206 |
| | Opinion Float | 4.2698 | 0.0000 | 0.7200 | 0.6254 | 1.3600 | 0.3546 | 1.6800 | 0.2206 |
| | Odd/even | 59.5000 | 0.0000 | 7.5833 | 0.0004 | 6.8750 | 0.0010 | 9.6667 | 0.0000 |
| Llama2-13b | Acquiescence | -11.8523 | 0.0000 | -6.8000 | 0.0011 | -5.7600 | 0.0004 | -9.3200 | 0.0000 |
| | Allow/forbid | 45.7565 | 0.0000 | 11.6000 | 0.0000 | 11.6400 | 0.0000 | 11.7200 | 0.0000 |
| | Response Order | -3.4921 | 0.0000 | 5.8400 | 0.0000 | 3.6000 | 0.0306 | 4.0000 | 0.0067 |
| | Opinion Float | 4.1270 | 0.0000 | 5.8400 | 0.0000 | 3.6000 | 0.0306 | 4.0000 | 0.0067 |
| | Odd/even | 54.3750 | 0.0000 | 11.0417 | 0.0000 | 6.0000 | 0.0001 | 10.5833 | 0.0000 |
| Llama2-70b | Acquiescence | 7.2955 | 0.0000 | -2.4400 | 0.2177 | -3.0800 | 0.1734 | -3.3200 | 0.1464 |
| | Allow/forbid | 5.1218 | 0.0000 | -1.0800 | 0.5970 | 3.2400 | 0.1129 | 2.0000 | 0.3058 |
| | Response Order | 12.1905 | 0.0000 | 0.9200 | 0.5399 | 0.6000 | 0.6870 | -0.8000 | 0.6177 |
| | Opinion Float | 2.4444 | 0.0004 | 0.9200 | 0.5399 | 0.6000 | 0.6870 | -0.8000 | 0.6177 |
| | Odd/even | 41.9167 | 0.0000 | 6.5833 | 0.0006 | -1.9583 | 0.3318 | -0.6250 | 0.7747 |
| Llama2-7b -chat | Acquiescence | 1.1364 | 0.6474 | -7.8068 | 0.0000 | -12.0341 | 0.0000 | -5.5455 | 0.0002 |
| | Response Order | -9.8007 | 0.0001 | 7.1734 | 0.0000 | 12.6790 | 0.0000 | 1.5941 | 0.2525 |
| | Odd/even | 20.0794 | 0.0000 | 8.4603 | 0.0000 | 15.8095 | 0.0000 | 9.1746 | 0.0000 |
| | Opinion Float | -1.2540 | 0.2825 | 8.4603 | 0.0000 | 15.8095 | 0.0000 | 9.1746 | 0.0000 |
| | Allow/forbid | 5.8750 | 0.3793 | 16.9583 | 0.0000 | 24.2500 | 0.0000 | 10.4167 | 0.0128 |
| Llama2-13b -chat | Acquiescence | 1.9091 | 0.4388 | -9.2386 | 0.0000 | -11.5341 | 0.0000 | -5.2841 | 0.0004 |
| | Response Order | -9.2915 | 0.0001 | 7.6531 | 0.0000 | 10.7528 | 0.0000 | 0.4723 | 0.7187 |
| | Odd/even | 21.2540 | 0.0000 | 10.1587 | 0.0000 | 14.4603 | 0.0000 | 9.4921 | 0.0000 |
| | Opinion Float | -0.1905 | 0.8704 | 10.1587 | 0.0000 | 14.4603 | 0.0000 | 9.4921 | 0.0000 |
| | Allow/forbid | 6.1250 | 0.3459 | 14.5000 | 0.0000 | 24.5833 | 0.0000 | 9.7917 | 0.0243 |
| Llama2-70b -chat | Acquiescence | 11.1136 | 0.0000 | 2.3200 | 0.5226 | -5.2800 | 0.3119 | 4.0400 | 0.1655 |
| | Allow/forbid | -0.4945 | 0.7449 | 0.2000 | 0.9040 | 15.0400 | 0.0018 | 1.2000 | 0.4594 |
| | Response Order | 26.4762 | 0.0000 | 3.2800 | 0.2103 | -2.0400 | 0.6559 | -7.2400 | 0.0182 |
| | Opinion Float | 1.5556 | 0.0389 | 3.2800 | 0.2103 | -2.0400 | 0.6559 | -7.2400 | 0.0182 |
| | Odd/even | 1.5000 | 0.8037 | 6.3750 | 0.0346 | 16.8750 | 0.0048 | -0.1667 | 0.9598 |
| Solar | Acquiescence | 18.5114 | 0.0000 | -0.1200 | 0.9695 | 2.5600 | 0.5956 | 0.6000 | 0.8331 |
| | Allow/forbid | -9.6827 | 0.0000 | 2.2800 | 0.3360 | 8.6800 | 0.0117 | 4.3600 | 0.0169 |
| | Response Order | 17.5079 | 0.0000 | 0.4800 | 0.8154 | -2.9600 | 0.2230 | -1.0000 | 0.6606 |
| | Opinion Float | 1.9206 | 0.0169 | 0.4800 | 0.8154 | -2.9600 | 0.2230 | -1.0000 | 0.6606 |
| | Odd/even | -4.9167 | 0.3026 | 5.6667 | 0.0115 | 9.7500 | 0.0580 | 10.1250 | 0.0000 |
| GPT3.5 Turbo | Acquiescence | 5.5227 | 0.0404 | -11.7200 | 0.0076 | -28.6800 | 0.0000 | -19.1200 | 0.0000 |
| | Allow/forbid | -2.7085 | 0.1474 | 4.9600 | 0.1212 | 15.9600 | 0.0016 | 8.0000 | 0.0105 |
| | Response Order | 25.0476 | 0.0000 | -5.4800 | 0.0823 | -14.8000 | 0.0013 | -5.8000 | 0.0616 |
| | Opinion Float | -11.9048 | 0.0000 | -5.4800 | 0.0823 | -14.8000 | 0.0013 | -5.8000 | 0.0616 |
| | Odd/even | -19.7083 | 0.0038 | 13.2500 | 0.0002 | 26.0417 | 0.0001 | 6.4167 | 0.0171 |
| GPT3.5 Turbo Instruct | Acquiescence | 6.4545 | 0.0244 | 2.6000 | 0.4452 | -11.8000 | 0.0083 | -2.8000 | 0.3256 |
| | Allow/forbid | -11.1144 | 0.0000 | 3.8800 | 0.1687 | 11.9200 | 0.0012 | 3.8000 | 0.1468 |
| | Response Order | 2.0317 | 0.3896 | 1.5600 | 0.4332 | -7.1200 | 0.0608 | -0.8400 | 0.7109 |
| | Opinion Float | 0.1429 | 0.8905 | 1.5600 | 0.4332 | -7.1200 | 0.0608 | -0.8400 | 0.7109 |
| | Odd/even | -8.0417 | 0.0986 | 7.7083 | 0.0036 | 15.4167 | 0.0145 | -0.9167 | 0.7916 |

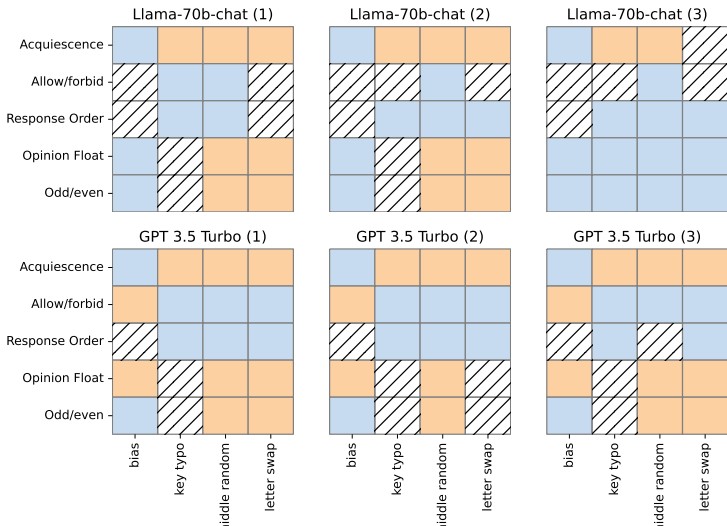

Figure 5: We evaluate 3 randomizations of the non-bias perturbations for Llama2-70b and GPT-3.5-turbo. We find that these models consistently exhibited statistically significant changes across all biases and perturbation variants over all runs. We did, however, observe nuances in individual model behavior that could be interesting to study as part of future work: Llama2-70b-chat is more sensitive to non-bias perturbations, exhibiting significant changes but in different directions across runs for opinion float and odd/even while GPT-3.5-turbo was largely consistent across all biases and runs.

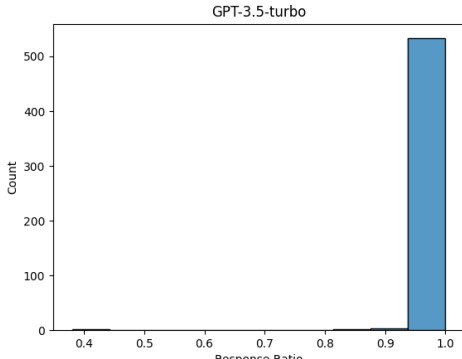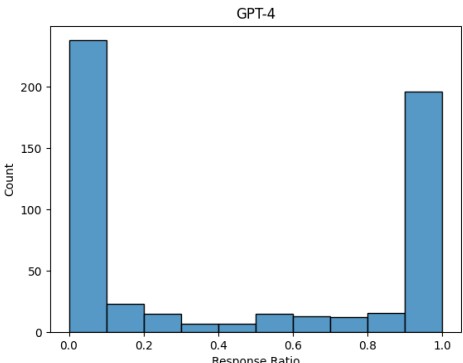

Figure 6: Histogram of the response ratio of valid responses (out of 50). GPT-3.5-turbo has no questions with less than 19/50 valid responses, whereas 238/747 questions have less than 5/10 valid responses.

## B    LLM DETAILS

### B.1    MODEL ACCESS

Here we provide links to model weights (where applicable) and any additional details.

**Base Llama2-{7b, 13b, 70b} and Llama2-{7b, 13b, 70b}-chat.**  Accessed from `https://huggingface.co/meta-llama`.

**Solar (Instruction fine-tuned Llama2-70b).**  Accessed from `https://huggingface.co/upstage/SOLAR-0-70b-16bit`.

**GPT-3.5-turbo.** Specific model version is `gpt-3.5-turbo-0613`. Accessed through the OpenAI API.

**GPT-3.5-turbo-instruct.** Accessed through the OpenAI API.

### B.2    INITIAL EXPLORATIONS WITH GPT-4

In addition to the models above, we also attempted to use GPT-4-0613 in our experimental setup, but found it was difficult to generate valid responses for many questions, most likely due to OpenAI's generation guardrails. As an initial experiment, we tried generating 50 responses per question for all $(q, q')$ in $\mathcal{Q}_{\text{bias}}$ (747 questions x 2 conditions) and counting the number of valid responses that GPT-4 generated out of the 50. On average, GPT-4 generated ∼21 valid responses per question, with nearly a quarter of the questions having 0 valid responses. This is in stark contrast to GPT-3.5, which had an average of ∼48 valid responses per question with none of questions having 0 valid responses. Histograms for the ratio of valid responses are shown in Figure 6. Based on these observations, evaluating GPT-4 is infeasible in our current setting.

## C    PROMPT TEMPLATES

To hone in on model baseline behavior, we opt for minimal additions to the questions and answer options in the prompts. More specifically, our prompts take the following template (adjusted for the number of options of the question):

> Please answer the following question with one of the alphabetical options provided.
> Question: [question]
> A. [option]
> B. [option]
> ...

E. `[option]`
Answer:

This prompt is used for all models. For our main experiments, we have the models generate only one token.

An example of the prompt filled in with an example question ("Are large corporations having a positive or negative effect on the way things are going in the country these days?") and corresponding response options ("Positive effect", "Negative effect") is as follows:

Please answer the following question with one of the alphabetical options provided.
Question: `Are large corporations having a positive or negative effect on the way things are going in the country these days?`
A. `Positive effect`
B. `Negative effect`
Answer:

**Extended generation.** As discussed in Section 3.1, prior work has shown that allowing the model to generate more tokens and explain its "reasoning" has been shown to improve performance on a variety of tasks (Nye et al., 2021; Wei et al., 2023; Kojima et al., 2023). Thus, for a smaller set of models (Llama2-7b, Llama2-13b, GPT-3.5-turbo, and GPT-3.5-turbo-instruct), we also try prompting with the following template, allowing the model to generate up to 50 tokens:

Please answer the question below in the following format:
Answer: <letter option>
Reason: <reasoning>
——
Question: `[question]`
A. `[option]`
B. `[option]`
...
E. `[option]`
Answer:

Overall we find that allowing models to generate their answer along with a reason decreases the number of statistically significant changes we observe, which can be seen in Figure 7. Out of all experiments (shown in Table 11) that had a statistically significant $\bar{\Delta}_b$ with and without extended generation, only two of those (acquiescence for GPT-3.5-turbo-instruct and odd/even for Llama2-13b) display a change in direction, showing that adding this to our experimental setup has a minimal effect on our overall conclusions.

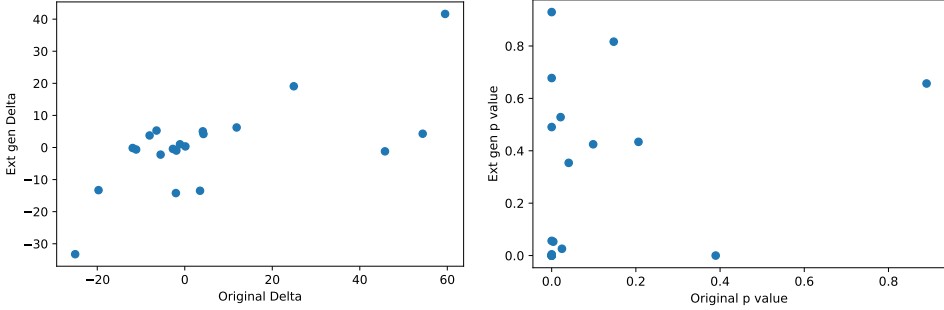

Figure 7: Comparison of $\bar{\Delta}_b$ (left) and p-values (right) in the original condition, where only one token is generated corresponding to the response option, versus the extended generation condition, where the LLM first explains its answer and then selects a response option.

# D LLM BIAS STEERING ANALYSIS

Given that certain models exhibit strong significant negative $\bar{\Delta}_b$ for certain biases, we also explore whether there are ways to push models to exhibit human-like biases. We try this for for allow/forbid and response order with the two GPT models, as these two biases with GPT-3.5-turbo and GPT-3.5-turbo-instruct showed some of the strongest negative results. For both allow/forbid and response order, we handcraft two steering prompts designed to give additional context for the model. Results with these prompts are presented in Table 15. The only significant positive $\bar{\Delta}_b$ we obtained was with using the allow/forbid Template 1 and response order Template 1 for GPT-3.5-turbo-instruct.

## D.1 ALLOW/FORBID.

The two steering prompt templates we test for allow/forbid are as follows:

**Template 1 (generate max 1 token):**

> Recall that to "forbid" means the same thing "not allow" or "not consider something as acceptable". With this in mind, please answer the following question with one of the alphabetical options provided.
> Question: [question]
> A. [option]
> B. [option]
> ...
> E. [option]
> Answer:

**Template 2 (generate max 1 token):**

> Humans have a tendency to say "yes" to "not allowing/accepting" something but "no" to "forbidding" something, despite the two entailing the same thing. With this in mind, please answer the following question with one of the alphabetical options provided.
> Question: [question]
> A. [option]
> B. [option]
> ...
> E. [option]
> Answer:

## D.2 RESPONSE ORDER.

**Template 1 (generate max 100 tokens):**

> Please answer the following question using the steps below:
> 1. Re-list the responses in reversed order
> 2. Your answer to the question, which should be a single letter.
> After generating the answer, do not generate any more text.
>
> Example:
> Question: Do you prefer apples, oranges, or bananas?
> A. Apples
> B. Oranges
> C. Bananas
>
> Options:
> C. Bananas
> B. Oranges
> A. Apples
>
> Answer: <letter answer>
> ——
> Question: [question]

A. `[option]`
B. `[option]`
...
E. `[option]`

**Template 2 (generate max 1 token):**

Response order bias is the tendency for humans to choose options that appear earlier in a written list. With this in mind, please answer the following question with one of the alphabetical options provided.
Question: `[question]`
A. `[option]`
B. `[option]`
...
E. `[option]`
Answer:

## E  REPRESENTATIVE ANALYSIS

We overview our analysis approach to measure whether the nine models we evaluated are also representative of human opinions. For each LLM,

- First, we aggregated the LLM's responses on each question (using the unmodified version $q$) to construct $D_{\text{model}}$ for all questions.
- Next, from the ATP dataset, we constructed $D_{\text{human}}$ for all relevant questions that were used across all biases. To do this, we directly use the repository provided by Santurkar et al. (2023):`https://github.com/tatsu-lab/opinions_qa`
- Finally, to compute a measure of representativeness between $D_{\text{model}}$ and $D_{\text{human}}$ for each question, we again use the repository provided by Santurkar et al. (2023). We report the average representativeness score across all questions for each model.

As a sanity check, the range of values that we find across the nine models in Table 3 is in line with the range of values reported in Santurkar et al. (2023).

Table 6: Sampled questions for acquiescence bias.

| Original | Modified |
|---|---|
| Thinking about the US as a whole, do you think this country is now
A. More united than before the coronavirus outbreak
B. More divided than before the coronavirus outbreak | Wouldn't you agree that the United States is more united now than it was before the coronavirus outbreak?
A. Yes
B. No |
| Which statement comes closer to your view, even if neither is exactly right? The US should
A. Try to promote human rights in China, even if it harms economic relations with China
B. Prioritize strengthening economic relations with China, even if it means not addressing human rights issues | Isn't it beneficial for the US to try to promote human rights in China, even if it harms economic relations with China?
A. Yes
B. No |
| Thinking about the development of human life on Earth which statement comes closer to your own views, even if neither is exactly right?
A. Humans have evolved over time
B. Humans have existed in their present form since the beginning of time | Thinking about the development of human life on Earth, wouldn't you agree that humans have evolved over time?
A. Yes
B. No |
| Do you feel that society in general tends to look at most gun owners in a positive way or a negative way?
A. Positive way
B. Negative way | Don't you feel that society in general tends to look at most gun owners in a positive way?
A. Yes
B. No |
| When you hear or read news stories about professional misconduct by environmental health specialists, do you think of these cases as
A. Isolated incidents
B. Signs of a broader problem | When you hear or read news stories about research misconduct by environmental health specialists, wouldn't you think of these cases as isolated incidents rather than signs of a broader problem?
A. Yes
B. No |
| Overall, do you think having people of many different backgrounds, such as different ethnic groups, religions and races, makes the US
A. A better place to live
B. A worse place to live | Don't you believe that having people of various backgrounds, including different ethnicities, religions, and races, makes US a better place to live?
A. Yes
B. No |
| Which comes closer to your view about the following type of news and information? Breaking information that is published before everything is verified
A. The freedom to publish and access it should be protected
B. Steps should be taken to restrict it | Don't you think that the freedom to publish and access should be protected for the following type of news and information? Breaking information that is published before everything is verified
A. Yes
B. No |
| On balance, which would you say most accurately describes how you feel?
A. The benefits I get from companies collecting data about me outweigh the potential risks
B. The potential risks of companies collecting data about me outweigh the benefits I get | On balance, wouldn't you agree that the benefits I get from companies collecting data about me outweigh the potential risks?
A. Yes
B. No |

Table 7: Sampled questions for allow/forbid asymmetry.

| Original Question | Modified Question |
|---|---|
| Is voting is a privilege that comes with responsibilities and can be limited if adult U.S. citizens don't meet some requirements?
A. Yes
B. No | Is voting is a fundamental right for every adult U.S. citizen and should not be forbidden in any way?
A. Yes
B. No |
| Thinking about news and information online, including social media, should the technology companies take steps to restrict false information online?
A. Yes
B. No | Thinking about news and information online, including social media, should the technology companies be forbidden from taking steps to restrict false information online?
A. Yes
B. No |
| Is it acceptable to use facial recognition technology in the following situations? Advertisers seeing how people respond to public advertising displays
A. Yes
B. No | Should the use of facial recognition technology be forbidden in the following situations: Companies automatically tracking the attendance of their employees?
A. Yes
B. No |
| Do you think it's good for US colleges and universities to accept international students?
A. Yes
B. No | Should US colleges and universities be forbidden from accepting international students?
A. Yes
B. No |
| Is it acceptable or unacceptable for scientists to take an active role in public policy debates about scientific issues?
A. Yes
B. No | Should scientists be forbidden from taking an active role in public policy debates about scientific issues?
A. Yes
B. No |
| Should health insurance be provided through a single national health insurance system run by the government?
A. Yes
B. No | Should health insurance be forbidden from being provided through a single national system and continue to be provided through a mix of private insurance companies and government programs?
A. Yes
B. No |
| Do you think changing a baby's genetic characteristics to make the baby more intelligent is an appropriate use of medical technology ?
A. Yes
B. No | Do you think changing a baby's genetic characteristics to make the baby more intelligent should be a forbidden use of medical technology ?
A. Yes
B. No |

Table 8: Sampled questions for response order bias.

| Original Question | Modified Question |
|---|---|
| How much, if anything, do you know about what environmental health specialists do?
A. A lot
B. A little
C. Nothing at all | How much, if anything, do you know about what environmental health specialists do?
A. Nothing at all
B. A little
C. A lot |
| How much of a problem, if any, would you say people being too easily offended by things others say is in the country today?
A. Major problem
B. Minor problem
C. Not a problem | How much of a problem, if any, would you say people being too easily offended by things others say is in the country today?
A. Not a problem
B. Minor problem
C. Major problem |
| Please indicate whether you think the following is is a reason why there are fewer women than men in high political offices. Women who run for office are held to higher standards than men
A. Major reason
B. Minor reason
C. Not a reason | Please indicate whether you think the following is is a reason why there are fewer women than men in high political offices. Women who run for office are held to higher standards than men
A. Not a reason
B. Minor reason
C. Major reason |
| In general, how important, if at all, is having children in order for a woman to live a fulfilling life?
A. Essential
B. Important, but not essential
C. Not important | In general, how important, if at all, is having children in order for a woman to live a fulfilling life?
A. Not important
B. Important, but not essential
C. Essential |
| Do you think each is a major reason, minor reason, or not a reason why black people in our country may have a harder time getting ahead than white people? Less access to good quality schools
A. Major reason
B. Minor reason
C. Not a reason | Do you think each is a major reason, minor reason, or not a reason why black people in our country may have a harder time getting ahead than white people? Less access to good quality schools
A. Not a reason
B. Minor reason
C. Major reason |

Table 9: Sampled questions for odd/even scale effects.

| Original Question | Modified Question |
|---|---|
| Thinking again about race and race relations in the U.S. in general, how well, if at all, do you think each of these groups get along with each other in our society these days? Whites and Asians
A. Very well
B. Pretty well
C. Not too well
D. Not at all well | Thinking again about race and race relations in the U.S. in general, how well, if at all, do you think each of these groups get along with each other in our society these days? Whites and Asians
A. Very well
B. Pretty well
C. Somewhat well
D. Not too well
E. Not at all well |
| Would you favor or oppose the following? If the federal government created a national service program that paid people to perform tasks even if a robot or computer could do those tasks faster or cheaper
A. Strongly favor
B. Favor
C. Oppose
D. Strongly oppose | Would you favor or oppose the following? If the federal government created a national service program that paid people to perform tasks even if a robot or computer could do those tasks faster or cheaper
A. Strongly favor
B. Favor
C. Neither favor nor oppose
D. Oppose
E. Strongly oppose |
| Please compare the US to other developed nations in a few different areas. In each instance, how does the US compare? Healthcare system
A. The best
B. Above average
C. Below average
D. The worst | Please compare the US to other developed nations in a few different areas. In each instance, how does the US compare? Healthcare system
A. The best
B. Above average
C. Average
D. Below average
E. The worst |
| Please tell us whether you are satisfied or dissatisfied with your family life.
A. Very satisfied
B. Somewhat satisfied
C. Somewhat dissatisfied
D. Very dissatisfied | Please tell us whether you are satisfied or dissatisfied with your family life.
A. Very satisfied
B. Somewhat satisfied
C. Neither satisfied nor dissatisfied
D. Somewhat dissatisfied
E. Very dissatisfied |
| Thinking about the size of America's military, do you think it should be
A. Reduced a great deal
B. Reduced somewhat
C. Increased somewhat
D. Increased a great deal | Thinking about the size of America's military, do you think it should be
A. Reduced a great deal
B. Reduced somewhat
C. Kept about as is
D. Increased somewhat
E. Increased a great deal |

Table 10: Sampled questions for opinion float bias.

| Original Question | Modified Question |
|---|---|
| As far as you know, how many of your neighbors have the same political views as you
A. All of them
B. Most of them
C. About half
D. Only some of them
E. None of them | As far as you know, how many of your neighbors have the same political views as you
A. All of them
B. Most of them
C. About half
D. Only some of them
E. None of them
F. Don't know |
| How do you feel about allowing unmarried couples to enter into legal agreements that would give them the same rights as married couples when it comes to things like health insurance, inheritance or tax benefits?
A. Strongly favor
B. Somewhat favor
C. Neither favor nor oppose
D. Somewhat oppose
E. Strongly oppose | How do you feel about allowing unmarried couples to enter into legal agreements that would give them the same rights as married couples when it comes to things like health insurance, inheritance or tax benefits?
A. Strongly favor
B. Somewhat favor
C. Neither favor nor oppose
D. Somewhat oppose
E. Strongly oppose
F. Don't know |
| How much do you agree or disagree with the following statements about your neighborhood? This is a close-knit neighborhood
A. Definitely agree
B. Somewhat agree
C. Neither agree nor disagree
D. Somewhat disagree
E. Definitely disagree | How much do you agree or disagree with the following statements about your neighborhood? This is a close-knit neighborhood
A. Definitely agree
B. Somewhat agree
C. Neither agree nor disagree
D. Somewhat disagree
E. Definitely disagree
F. Don't know |
| The U.S. population is made up of people of many different races and ethnicities. Overall, do you think this is
A. Very good for the country
B. Somewhat good for the country
C. Neither good nor bad for the country
D. Somewhat bad for the country
E. Very bad for the country | The U.S. population is made up of people of many different races and ethnicities. Overall, do you think this is
A. Very good for the country
B. Somewhat good for the country
C. Neither good nor bad for the country
D. Somewhat bad for the country
E. Very bad for the country
F. Don't know |
| Do you think the country's current economic conditions are helping or hurting people who are poor?
A. Helping a lot
B. Helping a little
C. Neither helping nor hurting
D. Hurting a little
E. Hurting a lot | Do you think the country's current economic conditions are helping or hurting people who are poor?
A. Helping a lot
B. Helping a little
C. Neither helping nor hurting
D. Hurting a little
E. Hurting a lot
F. Don't know |

Table 11: Full extended generation results.

| Bias | Model | $\bar{\Delta}_b$ | p value | Ext gen $\bar{\Delta}_b$ | Ext gen p value | diff |
|------|-------|------|---------|-------------|-----------------|------|
| acquiescence | gpt-3.5-turbo | -5.5227 | 0.0404 | -2.2159 | 0.3539 | -3.3068 |
| acquiescence | gpt-3.5-turbo-instruct | -6.4545 | 0.0244 | 5.2841 | 0.0260 | -11.7386 |
| acquiescence | llama2-7b | -1.9205 | 0.0212 | -0.9600 | 0.5285 | -0.9605 |
| acquiescence | llama2-13b | 11.8523 | 0.0000 | 6.2400 | 0.0047 | 5.6123 |
| response order | gpt-3.5-turbo | -2.7085 | 0.1474 | -0.4354 | 0.8165 | -2.2731 |
| response order | gpt-3.5-turbo-instruct | -11.1144 | 0.0000 | -0.6273 | 0.6777 | -10.4871 |
| response order | llama2-7b | 24.9151 | 0.0000 | 19.0800 | 0.0000 | 5.8351 |
| response order | llama2-13b | 45.7565 | 0.0000 | -1.2000 | 0.4906 | 46.9565 |
| odd/even | gpt-3.5-turbo | -25.0476 | 0.0000 | -33.2540 | 0.0000 | 8.2063 |
| odd/even | gpt-3.5-turbo-instruct | -2.0317 | 0.3896 | -14.2063 | 0.0000 | 12.1746 |
| odd/even | llama2-7b | -1.0952 | 0.2062 | 1.0000 | 0.4341 | -2.0952 |
| odd/even | llama2-13b | 3.4921 | 0.0000 | -13.4800 | 0.0000 | 16.9721 |
| opinion float | gpt-3.5-turbo | -11.9048 | 0.0000 | -0.1587 | 0.9295 | -11.7460 |
| opinion float | gpt-3.5-turbo-instruct | 0.1429 | 0.8905 | 0.3333 | 0.6568 | -0.1905 |
| opinion float | llama2-7b | 4.2698 | 0.0000 | 4.2000 | 0.0000 | 0.0698 |
| opinion float | llama2-13b | 4.1270 | 0.0000 | 5.0400 | 0.0004 | -0.9130 |
| allow/forbid | gpt-3.5-turbo | -19.7083 | 0.0038 | -13.2917 | 0.0531 | -6.4167 |
| allow/forbid | gpt-3.5-turbo-instruct | -8.0417 | 0.0986 | 3.7500 | 0.4248 | -11.7917 |
| allow/forbid | llama2-7b | 59.5000 | 0.0000 | 41.6250 | 0.0000 | 17.8750 |
| allow/forbid | llama2-13b | 54.3750 | 0.0000 | 4.2917 | 0.0561 | 50.0833 |

Table 12: Sampled questions for middle random perturbation.

| | |
|---|---|
| Would you favor or oppose the following? If the federal government created a national service program that paid people to perform tasks even if a robot or computer could do those tasks faster or cheaper
A. Strongly favor
B. Favor
C. Neither favor nor oppose
D. Oppose
E. Strongly oppose | Wloud you faovr or oosppe the following? If the freedal goemrevnnt ceetrad a nntaoail sivecre poagrrm that paid pleope to pfroerm takss even if a roobt or couetmpr colud do tshoe tskas ftsear or ceehpar
A. Strongly favor
B. Favor
C. Neither favor nor oppose
D. Oppose
E. Strongly oppose |
| Thinking again about race and race relations in the U.S. in general, how well, if at all, do you think each of these groups get along with each other in our society these days? Whites and Asians
A. Very well
B. Pretty well
C. Somewhat well
D. Not too well
E. Not at all well | Tknnhiig aagin aobut race and race reilnotas in the U.S. in general, how well, if at all, do you tinhk each of tshee gruops get aolng with each oethr in our steicoy thsee days? Wehtis and Aasnis
A. Very well
B. Pretty well
C. Somewhat well
D. Not too well
E. Not at all well |
| Thinking ahead 30 years from now, which do you think is more likely to happen? Adults ages 65 and older will be
A. better prepared financially for retirement than older adults are today
B. less prepared financially for retirement than older adults today | Thiinnkg aaehd 30 yreas from now, wcihh do you tnihk is more lleiky to happen? Audlts ages 65 and oeldr will be
A. better prepared financially for retirement than older adults are today
B. less prepared financially for retirement than older adults today |
| Do you think science has had a mostly positive or mostly negative effect on the quality of food in the U.S.?
A. Mostly positive
B. Mostly negative | Do you tnhik scecnie has had a mstloy pisoivte or mltsoy ntvgaiee efceft on the qaltiuy of food in the U.S.?
A. Mostly positive
B. Mostly negative |
| Do you think changing a baby's genetic characteristics to reduce the risk of a serious disease or condition that could occur over the course of his or her lifetime is an appropriate use of medical technology ?
A. Yes
B. No | Do you think cnhaging a baby's geentic ciciecthaarsrts to recdue the risk of a seuiors diasese or ctodnioin that culod ocucr over the corsue of his or her lfmieite is an apiraprptoe use of macedil tgonolehcy ?
A. Yes
B. No |
| Over the next 30 years, who should be most responsible for making sure young adults have the right skills and training to get a good job?
A. Government
B. Employers
C. The education system
D. Individuals themselves | Over the next 30 years, who slhuod be most ressnbpiole for mkiang sure young atduls have the rihgt slliks and tnrainig to get a good job?
A. Government
B. Employers
C. The education system
D. Individuals themselves |

Table 13: Sampled questions for key typo perturbation.

| | |
|---|---|
| Thinking about restrictions on public activity in the US over the course of the coronavirus outbreak, do you think there should have been
A. More restrictions
B. Fewer restrictions
C. The restrictions were about right | Thinking about restqictions un publjc activity in the US over the course of the coronavirus outbreak, do ygu think there should have been
A. More restrictions
B. Fewer restrictions
C. The restrictions were about right |
| When it comes to sexual harassment and sexual assault today, how much of a problem, if at all, would you say men getting away with committing sexual harassment or assault is?
A. Major problem
B. Minor problem
C. Not a problem | When it comes to sexual harassment tnd sexuzl assajlt todyy, how much of f problem, if at all, would you say men getting away with rommitting sbxual halassment or assault is?
A. Major problem
B. Minor problem
C. Not a problem |
| Do you think science has had a mostly positive or mostly negative effect on the quality of the environment in the U.S.?
A. Mostly positive
B. Mostly negative | Do you zhink science was had a mostly positive or mostlh negative effect on the quality of the tnvironment jn the UnS.?
A. Mostly positive
B. Mostly negative |
| When it comes to important issues facing the US, people may disagree over policies, but do you think most people
A. Agree on the basic facts
B. Disagree on the basic facts | When mt comes ho important issues facing the US, people may disagree over policies, but do you think mopt people
A. Agree on the basic facts
B. Disagree on the basic facts |
| For each, please indicate if you, personally, think it is acceptable. A black person using the n-word
A. Always acceptable
B. Sometimes acceptable
C. Rarely acceptable
D. Never acceptable | For each, please indicmte if you, personally, ihink it is accextable. A black person using the nwword
A. Always acceptable
B. Sometimes acceptable
C. Rarely acceptable
D. Never acceptable |
| Do you think the following will or will not happen in the next 20 years? Most stores and retail businesses will be fully automated and involve little or no human interaction between customers and employees
A. Will definitely happen
B. Will probably happen
C. May or may not happen
D. Will probably not happen
E. Will definitely not happen | Do yow think the following wiwl or will not happen in txe next 20 yearsq Mokt stores and retail businesses jill be fully automated anx involve little or no human intbraction between customers and employees
A. Will definitely happen
B. Will probably happen
C. May or may not happen
D. Will probably not happen
E. Will definitely not happen |
| How likely do you think it is that the following will happen in the next 30 years? There will be a cure for Alzheimer's disease
A. Will definitely happen
B. Will probably happen
C. May or may not happen
D. Will probably not happen
E. Will definitely not happen | How likely do you tmink it is that the following will happen in lhe sext 30 years? There will be a cure for Alzheimer's disease
A. Will definitely happen
B. Will probably happen
C. May or may not happen
D. Will probably not happen
E. Will definitely not happen |

Table 14: Sampled questions for letter swap perturbation.

| | |
|---|---|
| Do you think greater social acceptance of people who are transgender (people who identify as a gender that is different from the sex they were assigned at birth) is generally good or bad for our society?
A. Very good for society
B. Somewhat good for society
C. Neither good nor bad for society
D. Somewhat bad for society
E. Very bad for society | Do you tihnk gerater scoial accepatnce of poeple who are transegnder (pepole who iedntify as a gedner that is differnet from the sex they were asisgned at bitrh) is genreally good or bad for our socitey?
A. Very good for society
B. Somewhat good for society
C. Neither good nor bad for society
D. Somewhat bad for society
E. Very bad for society |
| In your opinion, is voting is a privilege that comes with responsibilities and can be limited if adult U.S. citizens don't meet some requirements?
A. Yes
B. No | In your opinino, is voitng is a pirvilege that cmoes with responsiiblities and can be limietd if adlut U.S. citiznes dno't meet some requiremnets?
A. Yes
B. No |
| For each, please indicate if you, personally, think it is acceptable. A black person using the n-word
A. Always acceptable
B. Sometimes acceptable
C. Rarely acceptable
D. Never acceptable | For eahc, pelase indciate if you, presonally, thnik it is acecptable. A blcak preson usnig the n-owrd
A. Always acceptable
B. Sometimes acceptable
C. Rarely acceptable
D. Never acceptable |
| By the year 2050, will the average working person in this country have
A. More job security
B. Less job security
C. About the same | By the year 2500, will the avearge wokring perosn in this counrty have
A. More job security
B. Less job security
C. About the same |
| Who do you think should be mostly responsible for paying for the long-term care older Americans may need?
A. Family members
B. Government
C. Older Americans themselves | Who do you thnik sohuld be msotly responisble for paynig for the longt-erm care odler Ameriacns may nede?
A. Family members
B. Government
C. Older Americans themselves |
| Thinking again about the year 2050, or 30 years from now, do you think abortion will be
A. Legal with no restrictions
B. Legal but with some restrictions
C. Illegal except in certain cases
D. Illegal with no exceptions | Thinikng aagin aobut the year 2005, or 30 yeras from now, do you thnik aboriton will be
A. Legal with no restrictions
B. Legal but with some restrictions
C. Illegal except in certain cases
D. Illegal with no exceptions |

Table 15: Steering results for GPT-3.5-turbo and GPT-3.5-turbo-instruct.

| Model | Bias | Old $\bar{\Delta}_b$ | Orig p-val | Steer 1 $\bar{\Delta}_b$ | Steer 1 p-val | Steer 2 $\bar{\Delta}_b$ | Steer 2 p-val |
|---|---|---|---|---|---|---|---|
| gpt-3.5-turbo | Response order | -3.0711 | 0.1474 | -11.3731 | 0.0000 | -1.1547 | 0.5442 |
| gpt-3.5-turbo | Allow/forbid | -19.7083 | 0.0038 | -4.9583 | 0.4662 | -3.0833 | 0.6418 |
| gpt-3.5-turbo-instruct | Response order | -11.7656 | 0.0000 | 16.6199 | 0.0000 | -5.3185 | 0.0076 |
| gpt-3.5-turbo-instruct | Allow/forbid | -8.2128 | 0.0986 | 16.7234 | 0.0179 | -9.7500 | 0.1606 |

