# OpenReview forum: "Do LLMs exhibit human-like response biases? A case study in survey design"
_ICLR.cc/2024/Conference — Submitted to ICLR 2024_

### Official Review · Reviewer_6vwz · 2023-10-31

**Soundness:** 2 fair
**Presentation:** 3 good
**Contribution:** 3 good
**Rating:** 5
**Confidence:** 3

**Summary:**

This paper studies LLMs as potential tools for simulating human behavior in the context of behavioral research. Motivated by previous research showing that human responses can be sensitive to changes in a prompt (e.g., phrasing and format), the paper seeks to explore to what extent LLMs are sensitive to such changes in prompts in that context. To do so the authors present BiasMonkey, a framework consisting of a dataset generation, an LLM response collection, and an LLM response evaluation phase. Dataset collection is achieved by modifying existing questions with the intention to induce response biases (e.g., response order, allow-forbid asymmetry), and also with perturbations that are not bias-inducing (e.g,, letter swap and adding typos) for human respondents. Evaluation is then conducted by comparing the response behavior between an original question and its perturbed counterpart. The authors experiment with Pew Research’s American Trends Panel (ATP) set of questions using 7 models based on Llama-2 and GPT-3.5. Experimental results demonstrate that only Llama-2-70B exhibits human-like behavior with respect to directional changes after perturbing prompts. However, the results on non-bias perturbations show that models also change their responses when prompted with such modified questions. Interestingly, the authors demonstrate that chain-of-thought prompting (i.e., asking the model for reasoning about its decision) yields less significant results.

**Strengths:**

* The paper discusses potential complications with using LLMs as proxies for human responses in behavioral research studies. Since this is a growing topic of interest in the community, it is important to address such potential shortcomings as early as possible. As such, I believe that this work represents an important contribution.
* The paper provides a detailed discussion section which includes potential implications and limitations of the obtained results.

**Weaknesses:**

* The response biases used in this work have not been verified with human crowdworkers on the dataset used for the LLM experiments. Instead, the work assumes the existence of such response biases based on existing works. To fully validate the experimental setup, it would be desirable to collect human responses for original and modified questions in the used dataset, and then compare the relative differences between humans and LLMs in response behavior.
* The paper makes an argument that statistically significant results demonstrated in this work may be a result of spurious correlations in the data. It would have been interesting for the authors to discuss this issue further, and potentially investigate this in more detail. Additional experiments with other datasets/sets of questions could potentially give clarity on that as well.
* The paper provides a direct comparison between LLM and human responses in section 3.3 (using Llama-2-70B). However, the section lacks critical details of the experimental setup. Which dataset and human responses were used for the comparison? Were responses collected specifically for that study, or obtained from an existing collection effort? Such details are imperative for the reader to understand the details of the reported results.

**Questions:**

* How robust are the model changes with non-bias perturbations? Did you experiment with different runs for random perturbations, to see if such changes persisted with different configurations? It would be interesting to see the variance here.

---

> ### Author Response · Authors · 2023-11-16
> **Response to Reviewer 6vwz**
>
> Thank you for your thorough review, we respond to the weaknesses and questions:
>
> *“The response biases used in this work have not been verified with human crowdworkers.”*: We agree that having parallel human data would be good to include! We considered running a human study on the same sets of questions, but upon further consideration found that a study at this scale would be very challenging for multiple reasons. First, we are primarily concerned about the potential sources of errors with collecting this data from crowdsourcing platforms like Mechanical Turk or Prolific. Unlike tasks with objective labels, it would be incredibly difficult to verify users’ faithfulness to the instructions as there is no notion of “correctness'' for these questions. Furthermore, it would be difficult to control for sources of sampling bias and the potential for lack of sufficient demographic diversity across respondents. As a result, we believe that a human study of this magnitude should stand as its own, separate work. We are discussing conducting a study with social scientists for such follow-up work.
>
> *“The paper makes an argument that statistically significant results demonstrated in this work may be a result of spurious correlations in the data.”*: In retrospect, we agree that our wording, which used the term “spurious correlations”, was imprecise and led to more questions than it provided answers to. We have amended this wording in Section 3.2: “Since LLMs are sensitive to both response biases and non-bias perturbations, it would be important to understand if there are common reasons underlying such behavior.” Investigating this behavior rigorously would likely require a more narrowed scope of questions, biases, and models, which we believe would be interesting future work.
>
> *“Which dataset and human responses were used for the comparison?”*: The human responses were obtained from a prior collection effort by Pew, as mentioned in our response to the reviewer’s first weakness. This set of human responses was also utilized by Santurkar et al. to evaluate whether LLMs can reflect different human opinions (i.e., “representativeness”). Overall, we agree that the details could have been presented more clearly, even if we largely re-used the analysis protocol from Santurkar et al. to compute representativeness. For completeness, we have updated Section 3.3 and provided more experimental details in Appendix E.
>
> *“How robust are the model changes with non-bias perturbations?”*: This is an interesting suggestion. For two models, Llama2-70b-chat and GPT-3.5-turbo, we ran 2 more randomizations of the non-bias perturbations and found that our original conclusions about the lack of robustness of LLMs to non-bias perturbations remain the same. These models consistently exhibited statistically significant changes across all biases and perturbation variants over all runs. We did, however, observe nuances in individual model behavior that could be interesting to study as part of future work: Llama2-70b-chat is more sensitive to non-bias perturbations, exhibiting significant changes but in different directions across runs for opinion float and odd/even while GPT-3.5-turbo was largely consistent across all biases and runs. We have updated A.4 with this ablation.

---

> ### Author Response · Authors · 2023-11-20
>
> Thanks again for your feedback on our work! We hope that you’ve had the chance to consider our comments from your review. Are there any further questions or concerns we should discuss? If so, please let us know so we can address them before the end of the discussion period. If not, we would appreciate if you could raise your score to indicate that the concerns have been addressed.

---

> > ### Comment · Reviewer_6vwz · 2023-11-22
> > **Thank for addressing my comments**
> >
> > I'd like to thank the authors for addressing my comments and concerns, and I greatly appreciate the detail. I will keep my scores fixed since I still believe that the study design necessitates human annotations on the used datasets to effectively analyze human-like response biases in LLMs.

---

### Official Review · Reviewer_voCN · 2023-10-31

**Soundness:** 3 good
**Presentation:** 2 fair
**Contribution:** 2 fair
**Rating:** 5
**Confidence:** 4

**Summary:**

The paper investigates whether Large Language Models (LLMs) exhibit human-like response biases, especially when faced with changes in survey design or instruction wording. The authors developed a framework named BIASMONKEY to evaluate the behavior of LLMs in relation to known human biases. The framework involves creating datasets with both original and modified survey questions, collecting LLM responses, and then evaluating those responses. They found that most LLMs do not consistently display human-like response biases. Only one model, Llama2-70b, generally exhibited behavior resembling human biases.

**Strengths:**

The presented bias types offer an intriguing and comprehensive insight into the various sources of human bias in survey responses. Such a detailed categorization underscores the importance of carefully crafting surveys to minimize these biases and capture genuine opinions.

**Weaknesses:**

The primary concern with the study is its emphasis on the LLM's inability to simulate human-like survey patterns. While this observation matters, the BiasMonkey framework appears to hold greater importance. The paper seems to relegate this framework to a supporting role, rather than spotlighting its potential significance. I would suggest that the authors reconsider their presentation and study design to highlight more clearly the insights derived from BiasMonkey.

**Questions:**

It is not clear why the specific data and 7 models are picked. It is not clear why these choice are representative and generalized for universal findings. It would be better if the authors can clarify that.

---

> ### Author Response · Authors · 2023-11-16
> **Response to Reviewer voCN**
>
> We thank the reviewer for their feedback and address the weaknesses and questions below:
>
> *“The primary concern with the study is its emphasis on the LLM's inability to simulate human-like survey patterns.”*: We respectfully disagree with this concern as we believe focusing **our evaluation of LLM’s ability to reflect human-like response biases is a timely and important study with broad implications for the current state of the field**. As both reviewers 9LjM and 6vwz noted, “Many people these days are using LLMs as human proxies in studies, and this paper shows that this is not always a good idea” and “Since this is a growing topic of interest in the community, it is important to address such potential shortcomings as early as possible. As such, I believe that this work represents an important contribution.” Of course, we agree that BiasMonkey is a useful framework that we employed to gain these insights, which can also help others conduct similar evaluations on other response biases of interest. We treat both (1) the empirical observations of patterns of LLM behavior and (2) the creation of BiasMonkey as an extensible framework for further studies as important contributions of our work.
>
> *“I would suggest that the authors reconsider their presentation and study design to highlight more clearly the insights derived from BiasMonkey.”*: Thanks for the feedback! Please see our general response **point #3**.
>
> *“It is not clear why the specific data and 7 models are picked. It is not clear why these choice are representative and generalized for universal findings.”*: We elaborate on our choice of data in the general response **point #1**. Regarding the choice of models, we discussed this in the submission in Section 2.2: “We selected LLMs based on multiple axes of consideration: open-source versus commercial models, whether the model has been instruction fine-tuned, whether the model has undergone reinforcement learning with human feedback (RLHF), and the number of model parameters.” We have provided additional results in general response **point #2**. While it is impossible to guarantee empirically that our choices can generalize universally, as the reviewer is suggesting, we believe the data and set of models that we select are sufficiently representative of the existing literature and use of LLMs in practice. If there are any notable omissions that the reviewer would suggest, we’d also be happy to consider them!

---

> ### Author Response · Authors · 2023-11-20
>
> Thanks again for your feedback on our work! We hope that you’ve had the chance to consider our comments from your review. Are there any further questions or concerns we should discuss? If so, please let us know so we can address them before the end of the discussion period. If not, we would appreciate if you could raise your score to indicate that the concerns have been addressed.

---

> > ### Comment · Reviewer_voCN · 2023-11-22
> > **Thanks for the response!**
> >
> > Thanks author for the response! I have updated my score.

---

### Official Review · Reviewer_vQr1 · 2023-11-07

**Soundness:** 3 good
**Presentation:** 3 good
**Contribution:** 2 fair
**Rating:** 5
**Confidence:** 4

**Summary:**

Authors introduce a framework called BiasMonkey to evaluate if LLMs exhibit human-like behaviour (response bias) in survey questionnaires. They conclude that out of seven models evaluated, only Llama2-70b displays human-like responses. They also perform experiments on other perturbations that humans are robust to and find that certain models are more prone to such perturbations than others.

**Strengths:**

* Authors explore an interesting area in the field of language models. Bias in LMs are explored at lengths but mostly from the lens of gender. Studying how responses change with modifications in questions is a good way of understanding how well models can perform when used as human proxies.
* Authors do a good job of analysing the results across various axes like model training type, size, bias types etc
* The paper has been well written with appropriate visuals and tables wherever needed.

**Weaknesses:**

* Though the work is a good preliminary study, I believe a stronger contribution would be identifying patterns and correlations behind the behaviours.
* The questionnaire also feels limited as has been pointed by authors themselves.

**Questions:**

NA
Please refer to weaknesses

---

> ### Author Response · Authors · 2023-11-16
> **Response to Reviewer vQr1**
>
> We thank the reviewer for their time and feedback. We address their weaknesses below:
>
> *“A stronger contribution would be identifying patterns and correlations behind the behaviours.”*: We believe we have identified multiple interesting patterns and correlations, as we spell out in the general response **point #3**. If we have misunderstood what the reviewer is referring to as “patterns and correlations”, please let us know and we would be happy to follow up.
>
> *“The questionnaire feels limited”*: We elaborate on our choice of dataset in our general response **point #1**. If the reviewer has more concrete suggestions for us to address, please let us know.

---

> ### Author Response · Authors · 2023-11-20
>
> Thanks again for your feedback on our work! We hope that you’ve had the chance to consider our comments from your review. Are there any further questions or concerns we should discuss? If so, please let us know so we can address them before the end of the discussion period. If not, we would appreciate if you could raise your score to indicate that the concerns have been addressed.

---

### Official Review · Reviewer_9LjM · 2023-11-08

**Soundness:** 3 good
**Presentation:** 3 good
**Contribution:** 3 good
**Rating:** 8
**Confidence:** 4

**Summary:**

This paper researches the question whether LLMs exhibit human-like response-biases in survey questionnaires. This is a timely and important research question in a time where many researchers are using LLMs as proxies for humans (e.g. "Discovering language model behaviors ..", Perez et al. 2022). They propose a framework called BiasMonkey, which prescribes a way for generating question-pairs that can evaluate whether an LLM exhibits a certain bias. For example, humans show bias when the word "forbidden" is used, answering differently to *"Is it ever acceptable for an unmarried couple to live together"* and *"Should it be forbidden for an unmarried couple to live together"*. The authors also generate questions that evaluate whether LLMs show biases where humans don't, e.g. humans usually don't answer differently when letters are swapped, e.g. *"Overall, do you think science has made life easier or more difficult for most people?"* versus *"Ovearll, do you tihnk sicence has made life eaiser or more diffiuclt for most poeple?"*. They evaluate 7 LLMs with this of which some are instruction-tuned and/or RLHF-ed and others are only pretrained. The findings are that none of the models show biases similar to humans, except LLaMa-70B (base pretrained, not fine-tuned). Most other models that show statistically significant changes for control questions (indicating human-like bias) also show it for non-bias perturbations (like the letter swap), which means the result can be explained by spurious correlations. The conclusion is that we should be careful when using LLMs as human proxies, but LLaMa-70B's results are promising. The authors further validate this result by comparing models with human opinion as measured by wasserstein distance on US survey questions, finding that LLaMa-70B is most aligned.

**Strengths:**

This paper studies a very important question and the results in my opinion need to be published. Many people these days are using LLMs as human proxies in studies, and this paper shows that this is not always a good idea, as well as showing that some models can be more suited to this than others. The writing is excellent and polished, the presentation of the results are clear. The authors do extensive experiments with clear findings, and use different prompting methods to test whether their results still hold when the models are allowed to generate reasoning traces.

**Weaknesses:**

The following weaknesses are minor, I would recommend acceptance for this paper as it is in it's current form.


- It's a bit unfortunate that you decided not to evaluate GPT-4, as that one is the most used as a human proxy. It would be very interesting to me to see the same study done for GPT-4. I would not be surprised if GPT-4 would perform similarly to LLaMa-70B.
- It would be very useful to indicate significance in table 3 instead of separately in a table where you need to go to see which of the effects are significant in table 3, e.g. with asterisk or something
- You make several claims about instruction-tuned models without doing a controlled study comparing base models to instruction-tuned counterparts (there are more differences between ChatGPT and LLaMa-2 than instruction-tuning). I am relatively convinced the difference is due to instruction-tuning, also because you do have LLaMa-70B and LLaMa-70b-chat (only difference is instruction tuning), but an interesting addition or direction for future work would be to study the impact of instruction tuning on human-like response biases. For example, could it be that instruction-tuned models are further trained on higher-quality data without typos etc, making them less robust to these kind of examples. Could this also mean that other types of perturbations not involving badly formatted text that humans are not sensitive to would be less of a problem for models?

**Questions:**

Why do you use a temperature of 1 and not 0 if you also append an alphabetical letter to each option to get different results out of the model? I might be misunderstanding what it means to "append an alphabetical letter to the options". Usually, I'd imagine one would opt for a temperature of 0 to get less random results, then again I also understand the need to generate a distribution of answers.

Nits:
- Be consistent in how you write LLaMa

---

> ### Author Response · Authors · 2023-11-16
> **Response to Reviewer 9LjM**
>
> We thank the reviewer for their positive comments and helpful suggestions. We have responded to the weaknesses and question below:
>
> *“It would be very interesting to me to see the same study done for GPT-4.”*: Thank you for the feedback! We agree that this is an interesting thing to investigate, and so we tried adding GPT-4 (0613) to our study. Unfortunately, we found that it is much more difficult to generate valid responses, most likely due to OpenAI’s generation guardrails. Furthermore, it was nearly impossible to get valid responses to many questions. As an initial experiment, we tried generating 50 responses per question for all $(q,q’)$ in $\mathcal{Q}_\text{bias}$, a total of 1494 questions, and counting the number of valid responses (e.g., one of the letter options) that GPT-3.5-turbo vs GPT-4 generated out of the 50. On average, GPT-4 generated ~21 valid responses per question, with nearly a quarter of the questions having 0 valid responses. For example, GPT-4 tended to generate “As” or “This” (and when asked to generate more tokens, GPT-4 generated “As a language model” or “This is subjective”). This is in stark contrast to GPT-3.5-turbo, which had an average of ~48 valid responses per question, with none of the questions having 0 valid responses. Based on these observations, evaluating GPT-4 is infeasible in our current setting. We have since added these details to Appendix B2.
>
> *“It would be very useful to indicate significance in table 3.”*: In Table 2 (formerly Table 3), we use the colors of the cell to indicate significance based on a 0.05 cut-off. We have clarified the caption to more clearly reflect this distinction. For our analysis, we don’t believe it is necessary to indicate more granular levels of significance below 0.05, as is common practice in statistical analyses. We refer readers to the Appendix for the full results.
>
> *“You make several claims about instruction-tuned models without doing a controlled study comparing base models to instruction-tuned counterparts.”*: We agree that further results beyond Llama2-70b are of interest! We provided additional evidence in general response **point #2**, namely with the addition of Llama2-7b-chat and Llama2-13b-chat.
>
> *“Why do you use a temperature of 1 and not 0 if you also append an alphabetical letter to each option to get different results out of the model?”* We apologize for the lack of clarity around prompting and sampling LLM responses. Indeed, we use a temperature of 1 to elicit a distribution of answers across the 50 samples that we collect for each question. In regards to appending an alphabetical letter, this is how we turn a Pew question into a prompt for the LLM (see Appendix C for a concrete example). We have clarified these details in Section 2.1.

---

> > ### Comment · Reviewer_9LjM · 2023-11-20
> > **Thanks for the response**
> >
> > Thank you for the response! My questions have been adequately answered and I will keep my original score.

---

### Official Review · Reviewer_BoSQ · 2023-11-10

**Soundness:** 3 good
**Presentation:** 4 excellent
**Contribution:** 3 good
**Rating:** 8
**Confidence:** 4

**Summary:**

This paper assesses whether LLMs are sensitive to prompt wording variation in analogous ways to humans. It finds that all but one of seven LMs do not.

To achieve this, the authors propose a framework for generating datasets with questions pairs that do and do not exhibit particular biases to assess whether LM responses to it match those of humans. The framework also includes baseline pairs which are different but don't affect humans.

**Strengths:**

- This paper is well-written and careful in its arguments. It was a pleasure to read.
- It adds important evidence to the body of work exploring the ability of LMs to simulate humans in responding to surveys and multiple-choice questions. It is well-grounded in former work, showing in most cases where results from this paper agree with other papers, esp. on surprising findings like RLHF'd models being less representative. It also is useful to highlight that humans are biased by framing, in the context of LMs being criticized for requiring prompt engineering.
- I was glad to see this dataset contain baselines of non-bias perturbations to which humans are robust.

**Weaknesses:**

- I would be more compelled by the evidence presented here if the questions matched those presented to humans, but it seems to be the case that the questions are generated by a LM. This feels like a weird way to evaluate whether LMs fall prey to same biases that humans do, and the authors don't justify it. Was it because the original questions from social science/psychology/etc. were not available? Wouldn't you want to replicate the result in the distribution of these better-established question pairs than in question pairs where, as I understand it, half of the question pair is natural and the other is generated by a LM to be inflected by the framing which the bias is sensitive to?
- While I can't find overt problems--the methodology seems neat and sound to me--I think the evidence could be heavier. For example, why did you sample only 50 questions? Why only ATP?

**Questions:**

How did you decide not to use questions from original social science establishing the biases explored in this paper, and instead use ATP?

**Details Of Ethics Concerns:**

This work is highlighting the differences between human and LM responses, and it provides evidence that we shouldn't use LMs to reason about how humans might respond to surveys. If it were instead arguing the opposite, it would need ethics review, but I see evidence as dissuasive of replicating human behavior.

---

> ### Author Response · Authors · 2023-11-16
> **Response to Reviewer BoSQ**
>
> We thank the reviewer for their positive comments and feedback. We address the weaknesses and questions in their review below:
>
> *“It seems to be the case that the questions are generated by a LM.”*: We would like to clarify that the question modifications were *not* generated by a language model and apologize for the lack of explicit note of this on our end. The modified forms of the questions for each bias were generated by either modifying them manually ourselves (as was the case for acquiescence and allow/forbid) or by performing systematic modifications such as automatically appending an option, removing an option, or reversing the order of options (for odd/even, opinion float, and response order). We have added a note of this in Section 2.2. These manual and systematic modifications were made to reflect the types of modifications used in prior social science studies, as discussed in A.2.
>
> *“Why did you sample only 50 questions?”*: Due to budget and time constraints, we sampled 50 questions for each perturbation setting. We agree that a larger-scale analysis is beneficial, and have since run experiments across all questions (747 questions in total) for each perturbation modification. Figure 3 and Table 5 have been updated accordingly. After scaling up the number of samples, our conclusion is still that while Llama2-70b is more human-like than other models, the general trend is that LLMs do not respond to instruction changes as humans would.
>
> *“Why ATP?”*: Please see our general response **point #1**.

---

> ### Author Response · Authors · 2023-11-20
>
> Thanks again for your feedback on our work! We hope that you’ve had the chance to consider our comments from your review. Are there any further questions or concerns we should discuss? If so, please let us know so we can address them before the end of the discussion period.

---

> > ### Comment · Reviewer_BoSQ · 2023-11-21
> >
> > Thank you for the additional clarifications and experimental results! This makes the work stronger and improves the weaknesses of the paper somewhat, in my view.
> >
> > I am raising my score from 6 to 8 given these changes.

---

### Author Response · Authors · 2023-11-16
**General Response to Reviewers**

First, thank you to all reviewers for the comments on the paper! We appreciate their efforts to strengthen our work. We respond to related comments in a general post:

**Point #1: Justifying the choice of data (Reviewer BoSQ, vQr1, voCN)**

*Why ATP?* In Section 2.2 of the original submission, we detailed why we selected the American Trends Panel as the source of original questions: “it covers a diverse set of topics, has a substantial number of questions, and the related survey was conducted relatively recently”. This dataset has also been used as the primary source of data in prior work (e.g., Santurkar et al. 2023). Furthermore, using the ATP allows us to access the raw response data of respondents, enabling us to examine the correlation (or lack thereof) between the results of BiasMonkey and that of other existing metrics like representativeness (from Santurkar et al. 2023), which we discuss in Section 3.3. Since ATP covers a wide variety of topics and is also labeled with these topics, it also allows us to see if the topic of the question may influence behavior, as we explore in A.4.

*Why not use existing social science questions?* While there are numerous social science studies on the biases we examine, the sets of questions used in these studies are of a much smaller scale compared to ATP (in both number and topical domain). For instance, McClendon 1991’s study included 12 questions, McFarland 1984’s study included 4 questions, O’Muircheartaigh et al. 2001’s study included 21 questions, whereas we selected 747 questions from the ATP to span 5 different biases. Using existing social science questions would also require us to put together multiple datasets (since each of these biases was studied individually), and the topics they cover differ from study to study, which may introduce confounding factors. Finally, many of these studies were conducted pre-2000 and often used current events/topics from the time in their stimuli, many of which are no longer relevant. In contrast, the ATP was collected between 2014-2022.

**Point #2: Further evidence of the effect of instruction fine-tuning (Reviewer 9LjM, voCN)**

One of our key findings, as discussed in Section 3, was identifying a pattern between training schemes and behavior: instruction fine-tuned and RLHF-ed models exhibit less human-like behavior compared to “vanilla” LLMs. **We have since strengthened this correlation by running BiasMonkey on two additional models: Llama2-7b-chat and Llama2-13b-chat**. We observed similar patterns as with Llama2-70b-chat, which show less human-like responses to the bias modifications than their “vanilla” counterparts (Llama2-7b and Llama2-13b), which were included in our original submission. We have added these results to Section 3.

**Point #3: Patterns, correlations, and insights from our analysis (Reviewer vQr1, voCN)**

We summarize three primary findings from our evaluation using BiasMonkey, which we have added to the introduction to provide a concise takeaway from our study:

1. **LLMs are generally not reflective of human-like behavior**: All models showed behavior notably unlike humans such as (1) a significant change in the opposite direction as known human biases, or (2) a significant change to non-bias perturbations that humans are insensitive to. In particular, eight of the nine models that we evaluated failed to consistently reflect human-like behavior on the five response biases that we studied. (Table 2 and Figure 3)

2. **Instruction fine-tuning makes LLM behavior less human-like**: Instruction fine-tuned models are also more likely to exhibit significant changes as a result of non-bias perturbations, despite not exhibiting a significant change to the modifications meant to elicit response biases. (Table 2 and Figure 3)

3. **There is little correlation between exhibiting response biases and other desirable metrics**: While we also find that Llama2-70b can better replicate human opinion distributions, when comparing the remaining models, we find that the ability to replicate human opinion distributions is not indicative of how well an LLM reflects human behavior. (Table 3)

We believe these findings are rigorous as we evaluated over 9 models on 747 questions across both the bias modifications and 3 non-bias perturbations, where we collected 50 samples per variant of the question. In total, this amounts to sampling over 2.3 million responses across these models.

---

### Meta-Review · Area_Chair_vJba · 2023-12-03

**Metareview:**

This paper studies whether LLM exhibits human-like response biases. To do so, the authors proposes a framework, BiasMonkey, that can generate datasets, collect and evaluate LLM responses. The authors find that most LLM models do not exhibit human-like response biases, highlighting that we should be careful when using LLM as human proxies for behavioral studies.

Overall, all reviewers agree that the paper tackles a very important and timely research question. It is thoughtfully executed and well-written.

The main concern during the discussion is that the claim on whether LLM exhibits human-like biases is not supported by real human data. Instead, it assumes the existence of such human biases from prior works. This might be a concern since, for example, it is possible that humans do not exhibit such biases in the data used in the experiments. The paper would be much stronger if, quoting the reviewer comment from the discussion: "the claims made in this work would need to be backed by human parallel data, which should be used to statistically correlate response patterns between humans and LLMs on the introduced dataset and report corresponding effect sizes for each comparison."  This concern was echoed by even the more positive reviewer during the discussion phase, making them reluctant to endorse the paper's acceptance.

**Justification For Why Not Higher Score:**

The main concern like in the lack of human annotations to back up the claim on whether LLM exhibit human biases.

**Justification For Why Not Lower Score:**

N/A

---

### Decision · Program_Chairs · 2024-01-16

Reject